# Beyond the Average: Distributional Causal Inference under Imperfect Compliance

**Undral Byambadalai**
CyberAgent, Inc.,
Tokyo, Japan
undral21@gmail.com

**Tomu Hirata**
Databricks Japan, Inc.,
Tokyo, Japan
hirata@mi.t.u-tokyo.ac.jp

**Tatsushi Oka**
Keio University
Tokyo, Japan
tatsushi.oka@keio.jp

**Shota Yasui**
CyberAgent, Inc.,
Tokyo, Japan
yasui_shota@cyberagent.co.jp

## Abstract

We study the estimation of distributional treatment effects in randomized experiments with imperfect compliance. When participants do not adhere to their assigned treatments, we leverage treatment assignment as an instrumental variable to identify the local distributional treatment effect—the difference in outcome distributions between treatment and control groups for the subpopulation of compliers. We propose a regression-adjusted estimator based on a distribution regression framework with Neyman-orthogonal moment conditions, enabling robustness and flexibility with high-dimensional covariates. Our approach accommodates continuous, discrete, and mixed discrete-continuous outcomes, and applies under a broad class of covariate-adaptive randomization schemes, including stratified block designs and simple random sampling. We derive the estimator's asymptotic distribution and show that it achieves the semiparametric efficiency bound. Simulation results demonstrate favorable finite-sample performance, and we demonstrate the method's practical relevance in an application to the Oregon Health Insurance Experiment.

## 1 Introduction

Randomized experiments are a cornerstone of causal inference, widely employed in both academic research (Duflo et al., 2007) and industry settings (Kohavi et al., 2020). In practice, however, subjects often deviate from their assigned treatments, leading to imperfect compliance. When compliance is not guaranteed, estimating the causal effect for the entire population is generally not possible, without imposing additional assumptions. However, a standard approach to address this issue is to use the random assignment as an instrumental variable (IV). This strategy allows for identification of the causal effect of treatment for the subset of individuals who comply with their assignment—known as the local average treatment effect (LATE) (Imbens and Angrist, 1994)—without requiring assumptions about how individuals self-select into treatment.

To improve covariate balance between treatment and control groups, researchers often use covariate-adaptive randomization (CAR), which stratifies individuals based on key covariates before assigning treatments within each stratum. The CAR framework includes various designs, such as stratified block randomization and Efron's biased coin design (Imbens and Rubin, 2015), with simple random sampling as a special case.

39th Conference on Neural Information Processing Systems (NeurIPS 2025).

While much of the literature focuses on estimating the average effects, this summary measure can obscure important heterogeneity in treatment responses. In this paper, we study the estimation of *distributional treatment effects* in randomized experiments with covariate-adaptive randomization and noncompliance, focusing on the local distributional treatment effect (LDTE)—defined as the difference in counterfactual outcome distributions for compliers across treatment arms. By examining the entire distribution of outcomes, rather than just the mean, we aim to provide a more nuanced understanding of how treatments affect different segments of the population.

We propose a regression-adjusted estimator for LDTEs that leverages auxiliary covariates beyond stratum indicators to improve efficiency. Our setup accommodates heterogeneous assignment probabilities and heterogeneous treatment effects. Estimation proceeds via a distribution regression framework combined with Neyman-orthogonal moment conditions (Chernozhukov et al., 2018, 2022), which provide robustness to first-order estimation errors in high-dimensional or complex nuisance components. These nuisance functions—conditional distribution functions given pre-treatment covariates—are estimated using flexible machine learning methods, including random forests, neural networks, and gradient boosting. Incorporating cross-fitting further strengthens robustness against estimation errors.

Despite the growing body of work on CAR and noncompliance in experimental settings, methods that estimate distributional treatment effects in the presence of both CAR and noncompliance remain scarce. For instance, Jiang et al. (2023) address quantile treatment effects under full compliance, and Jiang et al. (2024) study average treatment effects under CAR with imperfect compliance. However, to our knowledge, there are no existing methods that integrate regression adjustment and IV techniques for estimating full outcome distributions under CAR and noncompliance. This paper addresses that gap and makes the following contributions:

1. We develop a regression-adjusted estimator for distributional treatment effects under CAR with noncompliance, applicable to continuous, discrete, and mixed discrete-continuous outcomes.

2. We derive the asymptotic distribution of the estimator under CAR, generalizing beyond the traditional i.i.d. framework in causal inference.

3. We establish the semiparametric efficiency bound for the LDTE under CAR and show that our estimator attains this bound.

4. We validate our approach through simulation studies and an empirical application to the Oregon Health Insurance Experiment, where only 58% of subjects complied with their treatment assignment.

The remainder of the paper is structured as follows. Section 2 reviews related literature. Section 3 describes the problem setup and identification strategy. Section 4 introduces the proposed estimation method. Section 5 presents the asymptotic properties of our estimator. Section 6 reports simulation and empirical results. Section 7 concludes. The Appendix includes notation, technical proofs, and additional experimental details.

## 2 Related Literature

**Distributional treatment effects** Distributional and quantile treatment effects provide a more comprehensive view of treatment impacts beyond average effects. The concept of QTE was first introduced by Doksum (1974) and Lehmann and D'Abrera (1975), and has since inspired a broad literature developing estimation and inference methods for distributional effects across econometrics, statistics, and machine learning. Notable contributions include Heckman et al. (1997); Imbens and Rubin (1997); Koenker (2005); Bitler et al. (2006); Athey and Imbens (2006); Firpo (2007); Chernozhukov et al. (2013); Koenker et al. (2017); Belloni et al. (2017); Callaway et al. (2018); Callaway and Li (2019); Chernozhukov et al. (2019); Ge et al. (2020); Park et al. (2021); Zhou et al. (2022); Gunsilius (2023); Kallus and Oprescu (2023), among others. Most of this work focuses on conditional distributional and quantile treatment effects. In contrast, Oka et al. (2025), Byambadalai et al. (2024), and Hirata et al. (2025) examine unconditional distributional effects, though their analyses are restricted to settings with simple random sampling and full compliance. Byambadalai et al. (2025) also examine unconditional distributional effects under covariate-adaptive randomization, but their framework likewise assumes full compliance.

**Instrumental variables estimation of distributional causal effects**   Instrumental variables have a long-standing role in identifying causal effects in the presence of confounding, either by relying on additional structural assumptions (Haavelmo, 1943; Angrist et al., 1996) or by enabling partial identification under weaker conditions (Manski, 1990; Balke and Pearl, 1997). A key development in the estimation of distributional effects is the instrumental variable quantile regression (IVQR) framework, which estimates quantile functions across the outcome distribution under the rank similarity assumption (Chernozhukov and Hansen, 2004, 2005, 2006; Kaido and Wüthrich, 2021). An alternative approach by Abadie et al. (2002) focuses on local QTEs for the complier subpopulation, under the monotonicity assumption—a setting also considered in our work. Frölich and Melly (2013) similarly estimate unconditional QTEs under endogeneity, assuming monotonicity. Wüthrich (2020) provide a detailed comparison between IVQR and local QTE models. Additionally, Abadie (2002) introduce a Kolmogorov–Smirnov-type test for comparing complier outcome distributions in randomized experiments. Other contributions addressing distributional and quantile causal effects using IV methods under assumptions different from ours include Chernozhukov et al. (2007); Horowitz and Lee (2007); Briseño Sanchez et al. (2020); Kook and Pfister (2024); Kallus et al. (2024); Chernozhukov et al. (2024), among others.

**Regression adjustment under covariate-adaptive randomization**   Regression adjustment using pre-treatment covariates to improve precision in average treatment effect (ATE) estimation has been extensively studied under simple random sampling (Fisher, 1932; Cochran, 1977; Yang and Tsiatis, 2001; Rosenbaum, 2002; Freedman, 2008b,a; Tsiatis et al., 2008; Rosenblum and Van Der Laan, 2010; Lin, 2013; Berk et al., 2013; Ding et al., 2019). Recent work extends this to covariate-adaptive randomization. Cytrynbaum (2024) derive optimal linear adjustments for stratified designs, and Rafi (2023) characterize the semiparametric efficiency bound for ATE estimation. Other contributions include covariate adjustment in matched-pair designs (Bai et al., 2024), general form of adjustment in biostatistics (Bannick et al., 2023; Tu et al., 2023), and methods for parameters defined by estimating equations (Wang et al., 2023). While most of these focus on ATEs under full compliance, Jiang et al. (2023) study regression adjustment for the QTE, and Jiang et al. (2024) extend these ideas to the local ATE with imperfect compliance. Our work builds on this rich literature by targeting distributional causal effects under covariate-adaptive randomization and noncompliance.

**Semiparametric estimation**   Our work builds on the semiparametric estimation literature, which focuses on estimating low-dimensional parameters in the presence of possibly infinite-dimensional nuisance components. Foundational contributions include Robinson (1988); Bickel et al. (1993); Newey (1994); Robins and Rotnitzky (1995), with more recent developments in high-dimensional and machine learning settings by Chernozhukov et al. (2018); Ichimura and Newey (2022), among others. We formulate our estimation problem using Neyman-orthogonal moment conditions (Neyman, 1959; Chernozhukov et al., 2022), which provide robustness to errors in the estimation of nuisance components.

## 3   Setup and Notation

We consider a randomized experiment with binary treatment employing covariate-adaptive randomization, where imperfect compliance creates a discrepancy between treatment assignment and actual treatment receipt. Let $Y$ denote the observed outcome of interest, $Z \in \{0,1\}$ the random assignment, and $D \in \{0,1\}$ the actual treatment received. Within the potential outcome framework (Rubin, 1974; Imbens and Rubin, 2015), we define $Y(1)$ and $Y(0)$ as potential outcomes under treatment status $D = 1$ and $D = 0$, respectively. Similarly, $D(1)$ and $D(0)$ represent potential treatment statuses under assignment $Z = 1$ and $Z = 0$. In this setup, random assignment $Z$ serves as an instrumental variable affecting treatment $D$, which subsequently influences outcome $Y$. The exclusion restriction holds, as instrument $Z$ affects outcome $Y$ only through treatment $D$. Hence, we can write the observed outcome and treatment as

$$Y = D \cdot Y(1) + (1 - D) \cdot Y(0) \quad \text{and} \quad D = Z \cdot D(1) + (1 - Z) \cdot D(0). \tag{1}$$

Furthermore, we consider a covariate-adaptive randomization (CAR) setup in which each participant is assigned to a stratum $S \in \mathcal{S} := \{1, \ldots, S\}$, with additional covariates $X \in \mathcal{X} \subset \mathbb{R}^{d_x}$ available. Strata are typically constructed based on certain baseline covariates, and we allow $S$ and $X$ be dependent. We let $\pi_z(s) := P(Z = z \mid S = s) \in (0, 1)$ be the target assignment probability for

treatment $z \in \{0, 1\}$ in stratum $s$ and let $p(s) := P(S = s) > 0$ be the stratum size. Figure 1 depicts the relationship between the variables.

**Pre-Experiment**                                    **Post-Experiment**

Figure 1: The relationship between the variables. Solid arrows ($\longrightarrow$) represent direct causal pathways, while dashed arrows ($--\rightarrow$) denote conditioning or derivation relationships rather than direct causality.

We observe a data $\{(Y_i, D_i, Z_i, S_i, X_i)\}_{i=1}^n$ with a sample size of $n$. For each stratum $s \in \mathcal{S}$, let $n(s) := \sum_{i=1}^n \mathbb{1}_{\{S_i=s\}}$ denote the number of observations in stratum $s$, and $n_z(s) := \sum_{i=1}^n \mathbb{1}_{\{Z_i=z, S_i=s\}}$ represent the number of observations receiving assignment $z \in \{0, 1\}$ in stratum $s$. Here, $\mathbb{1}_{\{\cdot\}}$ denotes the indicator function, which equals 1 if the condition inside is true and 0 otherwise. Then, define the following empirical estimates: $\widehat{\pi}_z(s) := n_z(s)/n(s)$ the estimated target assignment and $\widehat{p}(s) := n(s)/n$ the proportion of observations falling in stratum $s$. We impose the following assumptions on the data generating process and the treatment assignment mechanism.

**Assumption 3.1** (Data generating process and treatment assignment). We have

(i) $\left\{\left(Y_i(0), Y_i(1), D_i(0), D_i(1), S_i, X_i\right)\right\}_{i=1}^n$ are independent and identically distributed

(ii) $\left\{\left(Y_i(0), Y_i(1), D_i(0), D_i(1), X_i\right)\right\}_{i=1}^n \perp\!\!\!\perp \{Z_i\}_{i=1}^n \mid \{S_i\}_{i=1}^n$,

(iii) $\widehat{\pi}_z(s) = \pi_z(s) + o_p(1)$ for every $s \in \mathcal{S}$ and $z \in \{0, 1\}$.

(iv) $\mathbb{P}\left(D_i(1) \geq D_i(0)\right) = 1$.

Assumption 3.1 (i) allows for cross-sectional dependence among treatment statuses $\{Z_i\}_{i=1}^n$, thereby accomodating many covariate-adaptive randomization schemes. Assumption 3.1 (ii) states that the assignment is independent of potential outcomes, potential treatment choices and pre-treatment covariates conditional on strata. Assumption 3.1 (iii) states the assignment probabilities converge to the target assignment probabilities as sample size increases.

Common randomization schemes satisfying Assumption 3.1 (i) to (iii) include simple random sampling, stratified block randomization, biased-coin design (Efron, 1971), and adaptive biased-coin design (Wei, 1978). Assumption 3.1 (iv) says that there are no defiers in the population. This assumption is also called the monotonicity assumption in the literature, and is the key assumption that allows for the identification of the causal effect within a specific subpopulation, known as *compliers*.

To clarify this, we introduce the four treatment compliance types as defined by Angrist et al. (1996). Never-takers consistently avoid the treatment, with $D(1) = 0$ and $D(0) = 0$. Defiers exhibit behavior opposite to the intended assignment, receiving the treatment when not encouraged ($D(0) = 1$) and avoiding it when encouraged ($D(1) = 0$). Compliers follow the assigned treatment status, such that $D(1) = 1$ and $D(0) = 0$. Always-takers are individuals who receive the treatment regardless of the instrument assignment, i.e., $D(1) = 1$ and $D(0) = 1$. Note that these types are not directly observable by the researcher.

We are interested in the distributional effects of receiving the treatment. To that end, let the distribution function of potential outcomes be denoted by

$$\mathbb{F}_{Y(d)}(y) := \mathbb{P}\left(Y(d) \leq y\right) \text{ for } d \in \{0, 1\}, y \in \mathcal{Y}. \tag{2}$$

Analogous to the local average treatment effect (LATE) of Imbens and Angrist (1994), we define the *local distributional treatment effect* (LDTE) as the difference in the distribution functions of the potential outcomes among compliers:

for $y \in \mathcal{Y}$. Here, compliers (i.e., those with $D(1) > D(0)$) refer to individuals who receive the treatment if and only if they are assigned to it. The following lemma demonstrates that, under Assumption 3.1, a random assignment can be used to identify the distributional causal effect of receiving the treatment for this subgroup.

**Lemma 3.2** (Local distributional treatment effect). *Suppose Assumptions 3.1 holds. Then, the local distributional treatment effect can be expressed as, for $y \in \mathcal{Y}$,*

$$\beta(y) = \frac{\sum_{s=1}^{S} p(s) \cdot (\mathbb{E}[\mathbb{1}_{\{Y \leq y\}} \mid Z = 1, S = s] - \mathbb{E}[\mathbb{1}_{\{Y \leq y\}} \mid Z = 0, S = s])}{\sum_{s=1}^{S} p(s) \cdot (\mathbb{E}[D \mid Z = 1, S = s] - \mathbb{E}[D \mid Z = 0, S = s])}. \tag{3}$$

Our formulation in (3) builds upon and extends the approach of Abadie (2002) to accommodate covariate-adaptive randomization through stratum-specific weights. Both the numerator and the denominator are written as weighted averages across strata indexed by $s$, with weights given by the distribution $p(s)$.

The numerator in (3) can be interpreted as the *intent-to-treat (ITT) distributional effect*—that is, the difference in the distribution functions of the outcome $Y$ between treatment and control groups defined by the random assignment $Z$. Importantly, this reflects the effect of being assigned to treatment, not of actually receiving treatment. The denominator in (3) represents the *first stage* of the instrumental variable approach. It captures the effect of the assignment $Z$ on the probability of receiving the treatment $D$, conditional on stratum $S = s$, and then averages this across strata. The first stage quantifies the degree of compliance with the assignment and ensures that the instrument is relevant (i.e., affects treatment uptake). A non-zero first stage is necessary for the IV estimator to be well-defined and to identify the treatment effect for compliers. Thus, the LDTE is obtained by scaling the ITT distributional effect by the strength of the first stage. Notably, the denominator is constant in $y$, so the variation in $\beta(y)$ across values of $y \in \mathcal{Y}$ reflects changes in the distribution of outcomes, not in the compliance rate.

Lastly, we also define the *local probability treatment effect* (LPTE)

$$LPTE(y_j) := \mathbb{P}\big(y_{j-1} < Y(1) \leq y_j \mid D(1) > D(0)\big) - \mathbb{P}\big(y_{j-1} < Y(0) \leq y_j \mid D(1) > D(0)\big),$$

for each $j = 1, \ldots, J$, where $\mathcal{Y}_J := \{y_1, \cdots, y_J\} \subset \mathcal{Y}$ and $y_0 = -\infty$. The LPTE measures treatment-induced changes in the probability mass of the outcome distribution within each interval $(y_{j-1}, y_j]$, effectively comparing the "histograms" of potential outcomes for compliers. The theoretical results developed for the LDTE extend directly to the LPTE by substituting the indicator functions $\mathbb{1}_{\{Y(d) \leq y_j\}}$ with $\mathbb{1}_{\{y_{j-1} < Y(d) \leq y_j\}}$ for $d \in \{0, 1\}$ in all relevant expressions.

## 4 Estimation

We propose a regression-adjusted LDTE estimator for $\{\beta(y)\}_{y \in \mathcal{Y}}$ incorporating the additional covariates $X_i$. For notational convenience, we define the following terms. The conditional probability of treatment given the instrument, stratum, and covariates:

$$\eta_z(s, x) := \mathbb{E}[D \mid Z = z, S = s, X = x]. \tag{4}$$

The conditional distribution function of $Y$ given the instrument, stratum, and covariates:

$$\mu_z(y, s, x) := \mathbb{E}[\mathbb{1}_{\{Y \leq y\}} \mid Z = z, S = s, X = x] \text{ for } y \in \mathcal{Y}. \tag{5}$$

The estimators for these quantities are denoted by $\widehat{\mu}_z(y, s, x)$ and $\widehat{\eta}_z(s, x)$, respectively. Since $X_i$ may be a continuous variable, the estimation of $\widehat{\mu}_z(y, s, x)$ and $\widehat{\eta}_z(s, x)$ relies on nonparametric methods, such as logistic regression, random forests, and other flexible machine learning (ML) approaches. In covariate-adaptive randomized experiments, the target assignment probability for treatment $z \in \{0, 1\}$ for a given stratum $s$, denoted by $\pi_z(s)$, is typically known in advance or can be

consistently estimated using its sample analog, defined as $\widehat{\pi}_z(s) = n_z(s)/n(s)$. Then, our proposed estimator for the LDTE for $y \in \mathcal{Y}$ is given by

$$\widehat{\beta}(y) := \frac{\frac{1}{n}\sum_{i=1}^{n}(\Xi_{1,i}^{Y}(y) - \Xi_{0,i}^{Y}(y))}{\frac{1}{n}\sum_{i=1}^{n}(\Xi_{1,i}^{D} - \Xi_{0,i}^{D})}, \tag{6}$$

where

$$\Xi_{z,i}^{Y}(y) = \frac{\mathbb{1}_{\{Z_i=z\}} \cdot \left(\mathbb{1}_{\{Y_i \leq y\}} - \widehat{\mu}_z(y, S_i, X_i)\right)}{\widehat{\pi}_z(S_i)} + \widehat{\mu}_z(y, S_i, X_i), \tag{7}$$

$$\Xi_{z,i}^{D} = \frac{\mathbb{1}_{\{Z_i=z\}} \cdot \left(D_i - \widehat{\eta}_z(S_i, X_i)\right)}{\widehat{\pi}_z(S_i)} + \widehat{\eta}_z(S_i, X_i), \quad \text{for } z = 0, 1. \tag{8}$$

The estimator presented in (6) follows the structure of the well-known augmented inverse propensity weighting (AIPW) estimator, which relies on a doubly robust moment condition (Robins et al., 1994; Robins and Rotnitzky, 1995). This moment condition satisfies the Neyman orthogonality property (Chernozhukov et al., 2018, 2022), ensuring that the estimator is first-order insensitive to the estimation errors of the nuisance functions $(\mu_z(\cdot), \eta_z(\cdot))$. To further improve robustness, we incorporate cross-fitting with $L$ folds $(L > 1)$ as proposed by Chernozhukov et al. (2018). The complete estimation procedure is detailed in Algorithm 1. Setting the adjustment terms $\widehat{\mu}_z(\cdot)$ and $\widehat{\eta}_z(\cdot)$ to zero yields the empirical (unadjusted) estimator for the LDTE, obtained by replacing each component in (3) with its sample analog.

---

**Algorithm 1** ML Regression-Adjusted LDTE Estimator with Cross-Fitting

---

1: **Input:** Data $\{(Y_i, D_i, Z_i, X_i, S_i)\}_{i=1}^{n}$ partitioned into $L$ folds; supervised learning model $\mathcal{M}$
2: **Step 1:** Model training and prediction
3: **for all** (level $y \in \mathcal{Y}$, fold $\ell \in \{1, ..., L\}$, stratum $s \in \mathcal{S}$, instrument $z \in \{0, 1\}$) **do**
4:     Train model $\mathcal{M}$ on data with instrument $Z_i = z$ in stratum $S_i = s$, excluding fold $\ell$
5:     Obtain predictions $\widehat{\mu}_z(y, S_i, X_i)$ and $\widehat{\eta}_z(S_i, X_i)$ for observations in fold $\ell$ with $S_i = s$
6: **end for**
7: **Step 2:** Treatment effect estimation
8: **for all** $y \in \mathcal{Y}$ **do**
9:     Compute $\widehat{\beta}(y)$ according to equation (6)
10: **end for**
11: **Output:** Regression-adjusted estimator $\{\widehat{\beta}(y)\}_{y \in \mathcal{Y}}$

---

## 5  Asymptotic Properties

In this section, we derive the asymptotic distribution of our proposed estimator, which enables statistical inference and the construction of confidence intervals. Additionally, we establish the semiparametric efficiency bound for the LDTE and demonstrate that the regression-adjusted estimator achieves this bound under the specified assumptions. We begin by introducing some additional notation to formalize our results. Let $\ell^{\infty}(\mathcal{Y})$ be the space of uniformly bounded functions mapping an arbitrary index set $\mathcal{Y}$ to the real line.

**Assumption 5.1.** We have (i) For $z \in \{0, 1\}$ and $s \in \mathcal{S}$, define $I_z(s) := \{i \in [n] : Z_i = z, S_i = s\}$, $\delta_z^{Y}(y, s, X_i) := \widehat{\mu}_z(y, s, X_i) - \mu_z(y, s, X_i)$, and $\delta_z^{D}(s, X_i) := \widehat{\eta}_z(s, X_i) - \eta_z(s, X_i)$. Then, for $z \in \{0, 1\}$, we have

$$\sup_{y \in \mathcal{Y}, s \in \mathcal{S}} \left| \frac{\sum_{i \in I_1(s)} \delta_z^{Y}(y, s, X_i)}{n_1(s)} - \frac{\sum_{i \in I_0(s)} \delta_z^{Y}(y, s, X_i)}{n_0(s)} \right| = o_p(n^{-1/2}), \tag{9}$$

$$\max_{s \in \mathcal{S}} \left| \frac{\sum_{i \in I_1(s)} \delta_z^{D}(s, X_i)}{n_1(s)} - \frac{\sum_{i \in I_0(s)} \delta_z^{D}(s, X_i)}{n_0(s)} \right| = o_p(n^{-1/2}). \tag{10}$$

(ii) For $z \in \{0,1\}$, let $\mathcal{F}_z = \{\mu_z(y,s,x) : y \in \mathcal{Y}\}$ with an envelope $F_z(s,x)$. Then, $\max_{s \in \mathcal{S}} \mathbb{E}[|F_z(S_i, X_i)|^q | S_i = s] < \infty$ for $q > 2$ and there exist fixed constants $(\alpha, v) > 0$ such that

$$\sup_Q N\left(\varepsilon \|F_z\|_{Q,2}, \mathcal{F}_z, L_2(Q)\right) \leq \left(\frac{\alpha}{\varepsilon}\right)^v, \quad \forall \varepsilon \in (0,1], \tag{11}$$

where $N(\cdot)$ denotes the covering number and the supremum is taken over all finitely discrete probability measures $Q$.

Assumption 5.1(i) provides a high-level condition on the estimation of $\widehat{\mu}_z(y, s, X_i)$ and $\widehat{\eta}_z(s, X_i)$. Assumptions 5.1(ii) impose mild regularity condition on $\mu_z(y, s, X_i)$. Specifically, it holds automatically when $\mathcal{Y}$ is a finite set. We now present the weak convergence of our proposed estimator in the following theorem, which provides the theoretical foundation for conducting statistical inference. This asymptotic result enables the construction of confidence intervals using either sample-based estimates of the asymptotic variance or bootstrap methods. Further details on the inference procedure are provided in Appendix D.

We define $Y(D(z)) := D(z) \cdot Y(1) + (1 - D(z)) \cdot Y(0)$. With this notation, the observed outcome $Y$ can be expressed as $Y = Z \cdot Y(D(1)) + (1 - Z) \cdot Y(D(0))$. For $z \in \{0,1\}$, let $Y_i^z(y) := \mathbb{1}_{\{Y_i(D_i(z)) \leq y\}}$ and $\tilde{Y}_i^z(y) := Y_i^z(y) - \mathbb{E}[Y_i^z(y)|S_i]$. Also, let $\tilde{D}_i(z) := D_i(z) - \mathbb{E}[D_i(z)|S_i]$, $\tilde{\mu}_z(y, S_i, X_i) := \mu_z(y, S_i, X_i) - \mathbb{E}[\mu_z(y, S_i, X_i)|S_i]$ and $\tilde{\eta}_z(S_i, X_i) := \eta_z(S_i, X_i) - \mathbb{E}[\eta_z(S_i, X_i)|S_i]$ for $z \in \{0,1\}$. Then, we define

$$\phi_i(y,z) := \left(1 - \frac{1}{\pi_z(S_i)}\right)\tilde{\mu}_z(y, S_i, X_i) - \tilde{\mu}_{1-z}(y, S_i, X_i) + \frac{\tilde{Y}_i^z(y)}{\pi_z(S_i)}$$

$$- \beta(y)\left(\left(1 - \frac{1}{\pi_z(S_i)}\right)\tilde{\eta}_z(S_i, X_i) - \tilde{\eta}_{1-z}(S_i, X_i) + \frac{\tilde{D}_i(z)}{\pi_z(S_i)}\right) \text{ for } z \in \{0,1\}, \tag{12}$$

and

$$\xi_i(y) := \mathbb{E}[Y_i^1(y) - Y_i^0(y)|S_i] - \mathbb{E}[Y_i^1(y) - Y_i^0(y)]$$
$$- \beta(y)\left(\mathbb{E}[D_i(1) - D_i(0)|S_i] - \mathbb{E}[D_i(1) - D_i(0)]\right). \tag{13}$$

**Theorem 5.2** (Asymptotic Distribution). *Suppose Assumptions 3.1 and 5.1 hold. Then, in $\ell^\infty(\mathcal{Y})$, uniformly over $y \in \mathcal{Y}$, the regression-adjusted estimator defined in Algorithm 1 satisfies*

$$\sqrt{n}(\widehat{\beta}(y) - \beta(y)) \rightsquigarrow \mathcal{G}(y), \tag{14}$$

*where $\mathcal{G}(y)$ is a Gaussian process with covariance kernel*

$$\Omega(y, y') := \frac{\Omega_0(y, y') + \Omega_1(y, y') + \Omega_2(y, y')}{\mathbb{E}[D(1) - D(0)]^2}, \tag{15}$$

*with $\Omega_z(y, y') := \mathbb{E}[\pi_z(S_i)\phi_i(y, z)\phi_i(y', z)]$ for $z \in \{0,1\}$ and $\Omega_2(y, y') := \mathbb{E}[\xi_i(y)\xi_i(y')]$.*

We next derive the semiparametric efficiency bound of the LDTE and show our estimator achieves this bound in the following theorem. This implies that the asymptotic variance of any regular, root-$n$ consistent, and asymptotically normal estimator of the LDTE cannot be lower than this bound.

**Theorem 5.3** (Semiparametric Efficiency Bound). *Under Assumption 3.1, for every $y \in \mathcal{Y}$,*

(a) *the semiparametric efficiency bound for $\beta(y)$ is $\Omega(y)$, which is defined by*

$$\Omega(y) := \frac{\Omega_0(y, y) + \Omega_1(y, y) + \Omega_2(y, y)}{\mathbb{E}[D(1) - D(0)]^2}, \tag{16}$$

*where $\Omega_0(\cdot)$, $\Omega_1(\cdot)$ and $\Omega_2(\cdot)$ are defined in Theorem 5.2.*

(b) *furthermore if Assumption 5.1 also holds, then the regression-adjusted estimator $\widehat{\beta}(y)$ attains the semiparametric efficiency bound.*

As a corollary to the theorem above, the asymptotic variance of the regression-adjusted estimator with known nuisance functions is lower than that of the empirical (unadjusted) estimator, in which the adjustment terms are set to zero.

# 6 Experiments

## 6.1 Simulation Study

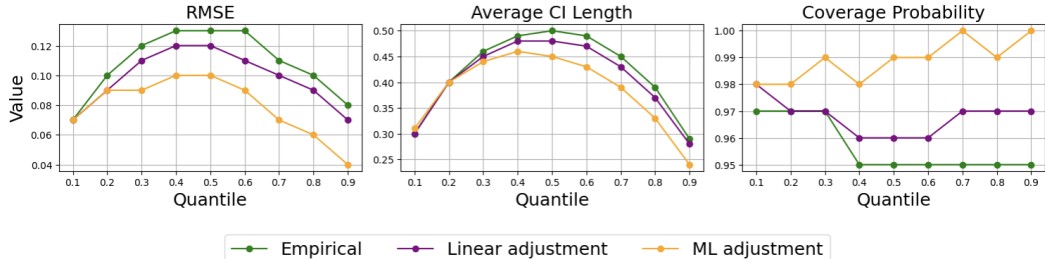

Figure 2: Simulation results for LDTE estimators under a nonlinear, high-dimensional design ($n = 1000$). RMSE, average 95% CI length, and empirical coverage are shown across quantiles $\{0.1, \ldots, 0.9\}$ based on 1000 replications. Three estimators are compared: unadjusted, linearly adjusted, and ML-adjusted (gradient boosting with 2-fold cross-fitting). Both adjusted estimators improve RMSE and CI length; the unadjusted estimator attains near-nominal coverage, while ML adjustment slightly over-covers, reflecting conservative inference.

We assess the finite-sample performance of our estimator through a simulation study designed to reflect a complex, nonlinear data-generating process with high-dimensional covariates and treatment effect heterogeneity.

The data generating process consists of four strata ($S = 4$) constructed by partitioning the support of a covariate $W_i \sim U(0, 1)$ into $S$ equal-length intervals, where $S_i$ indicates the interval containing $W_i$. For each unit $i$, we draw an additional 20-dimensional covariate vector $X_i = (X_{1,i}, \ldots, X_{20,i})^\top$ from a multivariate normal distribution $\mathcal{N}(0, I_{20 \times 20})$. The treatment indicator $Z_i$ follows a Bernoulli distribution with probability 0.5 within each stratum, maintaining a constant target proportion of treated units ($Z_i = 1$) across strata with $\pi_1(s) = 0.5$ for all $s \in \mathcal{S}$. The complete specification of the data-generating process is given by:

$$Y_i(d) = a_d + b(X_i, W_i) + \epsilon_i \quad \text{for } d \in \{0, 1\} \tag{17}$$

$$D_i(0) = \mathbb{1}_{\{b_0 + c(X_i, W_i) > c_0 \epsilon_i\}}, \tag{18}$$

$$D_i(1) = \begin{cases} \mathbb{1}_{\{b_1 + c(X_i, W_i) > c_1 \epsilon_i\}}, & \text{if } D_i(0) = 0, \\ 1, & \text{otherwise,} \end{cases} \tag{19}$$

where $(a_1, a_0, b_1, b_0, c_1, c_0) = (2, 1, 1, -1, 3, 3)$, and error term $\epsilon_i \sim \mathcal{N}(0, 1)$ with

$$b(X_i, W_i) = \sin(\pi X_{i1} X_{i2}) + 2(X_{i3} - 0.5)^2 + X_{i4} + 0.5 X_{i5} + 0.1 W_i, \tag{20}$$

$$c(X_i, W_i) = 0.1(X_{i1} + \log(1 + \exp(X_{i2})) + W_i). \tag{21}$$

This design incorporates nonlinear dependencies, integrates deliberately irrelevant covariates, and preserves the monotonicity assumption by eliminating the possibility of defiers.

We draw a sample of sizes $\{500, 1000, 5000\}$ from the data-generating process and estimate the LDTE at quantiles $\{0.1, ..., 0.9\}$ using three methods with 1000 simulations: an unadjusted estimator, a linear regression-adjusted estimator, and a machine learning-adjusted estimator based on gradient boosting. A reference sample of size $10^6$ is used to approximate ground-truth LDTE values. All adjusted estimators use 2-fold cross-fitting.

Figure 2 reports RMSE, average length and coverage of 95% confidence interval (CI) based on sample estimates. Both adjusted estimators achieve lower RMSE and shorter CIs than the unadjusted estimator. The unadjusted estimator achieves nominal 95% coverage for most quantiles, while ML adjustment exhibits slight over-coverage (up to 0.98–1.00), suggesting conservative intervals that could be tightened with improved nuisance estimation. Figure 3 shows RMSE reduction (%) relative to the unadjusted estimator. Linear adjustment yields modest gains (1–10%), while ML adjustment achieves up to 50% reduction for some quantiles, with performance improving as sample

size increases. These findings highlight the value of flexible regression adjustment in improving finite-sample efficiency for distributional causal effect estimation.

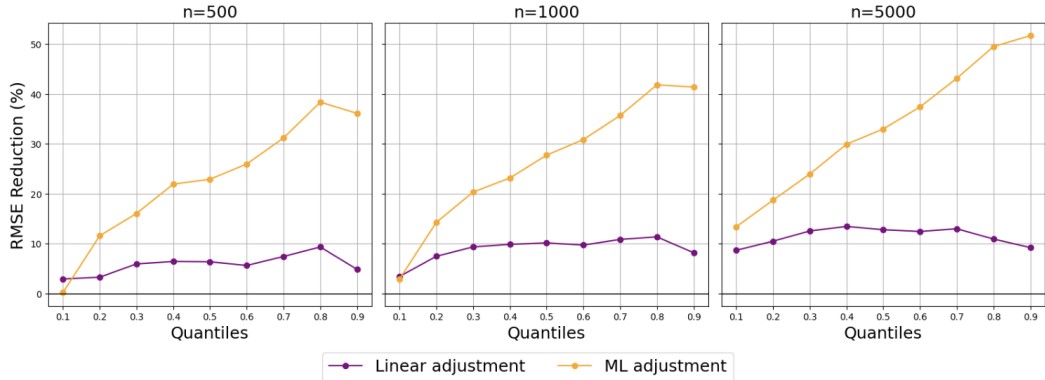

Figure 3: RMSE reduction (%) of adjusted estimators relative to the unadjusted estimator across quantiles and sample sizes. Linear adjustment yields modest efficiency gains (1–10%), while ML adjustment achieves up to 50% reduction, with improvements becoming more pronounced as sample size increases.

## 6.2 Real Data Analysis: Oregon Health Insurance Experiment

This subsection analyzes the impact of insurance coverage on emergency department (ED) visits using data from the Oregon Health Insurance Experiment.[1] We replicate the analysis in Finkelstein et al. (2016) and estimate distributional treatment effects. In 2008, the state of Oregon conducted a lottery to allocate health insurance to a group of uninsured low-income adults. Treatment assignment in this experiment was randomized based on household size, making the number of household members a stratification variable. However, due to imperfect compliance, not all individuals offered coverage enrolled, while some who were not selected obtained insurance through other means. Table 1 displays the sample breakdown by assigned and realized treatments, and only 58% of the subjects comply with their random assignment. For a detailed discussion of the experiment and average treatment effect estimates of insurance coverage on various other outcomes, see Finkelstein et al. (2012).

Table 1: Sample breakdown by assigned and realized treatments (sample counts and proportions)

| Realized treatment | Assigned treatment | | Total |
| | $Z = 0$ | $Z = 1$ | |
| --- | --- | --- | --- |
| $D = 0$ | 7596 (45%) | 6244 (37%) | 13840 (82%) |
| $D = 1$ | 910 (5%) | 2271 (13%) | 3181 (18 %) |
| Total | 8506 (50%) | 8515 (50%) | 17021 (100%) |

Figure 4 displays the distributional and probability treatment effect of insurance coverage on ED visits. We compute the LDTE and Local Probability Treatment Effect (LPTE) for $y \in \{0, 1, \ldots, 15\}$ accounting for the stratified design and imperfect compliance. For regression adjustment, we use gradient boosting with 5-fold cross-fitting, with 28 pre-treatment covariates ($X_i$) including various variables regarding past emergency department visits. The full list of covariates can be found in the Appendix.

The top-left panel of Figure 4 displays the empirical LDTE, while the top-right panel presents the regression-adjusted LDTE. Shaded areas represent 95% confidence bands, constructed using 500 bootstrap replications. In this case, regression adjustment reduces standard errors by approximately 0.5–15%. Similarly, the bottom-left panel shows the empirical LPTE, and the bottom-right panel shows the regression-adjusted LPTE. Here, the standard errors decrease by about 3.5–26.5% across

---

[1]The dataset is publicly available at https://www.nber.org/research/data/oregon-health-insurance-experiment-data.

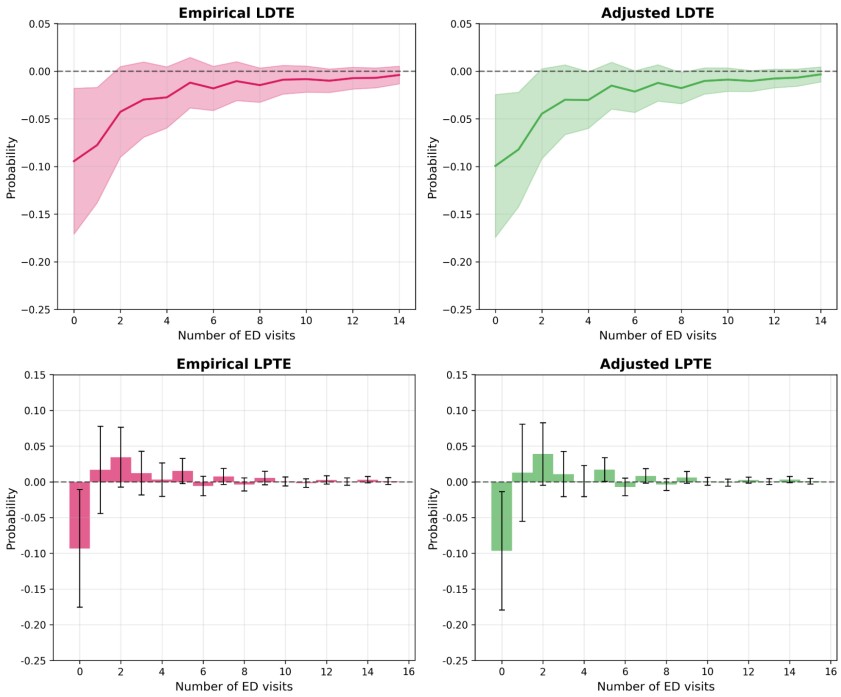

Figure 4: Oregon Health Insurance Experiment: Local Distributional Treatment Effect (LDTE) and Local Probability Treatment Effect (LPTE) of insurance coverage on number of emergency department (ED) visits. The left panels depict the empirical probability estimates, while the right panels present regression-adjusted estimates obtained using gradient boosting with 5-fold cross-fitting. Shaded regions and error bars represent 95% confidence intervals. Sample size: $n = 17{,}021$.

most of the distribution, except at $y \in \{0, 1, 2, 3\}$, where a slight increase in standard errors is observed.

The regression-adjusted distributional analysis reveals that the probability of having zero emergency department visits decreases by 9 percentage points (pp), with a standard error of 4.2 pp. Beyond this, the only marginally significant effect at the 5% level is an increase of approximately 1.7 pp in the probability of having five ED visits, with a standard error of 0.8 pp. No other statistically significant changes are observed across the rest of the distribution, even after applying regression adjustment.

## 7   Conclusion

We introduced a method for estimating local distributional treatment effects in randomized experiments with covariate-adaptive randomization and imperfect compliance. Our approach combines instrumental variable techniques with regression adjustment in a distribution regression framework, leveraging auxiliary covariates and modern machine learning for improved efficiency. The estimator is asymptotically normal, achieves the semiparametric efficiency bound, and performs well in simulations. We also demonstrated its practical relevance using data from the Oregon Health Insurance Experiment.

This work has several limitations. It relies on standard IV assumptions such as monotonicity and the exclusion restriction, and focuses on binary treatments. Performance may vary depending on the quality of nuisance estimation in finite samples. Future research could extend the framework to multi-valued or continuous treatments, relax identifying assumptions, and explore dynamic or longitudinal settings. Furthermore, extending the non-asymptotic frameworks developed by Su et al. (2023a,b) to a distributional setting represents a promising avenue for future research.

# 8 Acknowledgements

We are deeply grateful to the four anonymous reviewers and the program chairs for their thoughtful feedback and constructive discussions, which greatly improved the quality of this paper. Oka also acknowledges the financial support provided by JSPS KAKENHI (Grant Number 24K04821).

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

# Appendix

The Appendix is structured as follows. Section A provides a table summarizing the notation. Section B introduces some definitions. Section C presents all proofs. Section D discusses the construction of confidence intervals. Section E presents some additional experimental details.

## A  Summary of Notation

Table 2: Summary of Notation

| | |
|---|---|
| $X_i$ | pre-treatment covariates |
| $S_i$ | stratum indicator |
| $D_i$ | actual treatment received |
| $Z_i$ | treatment assignment |
| $Y_i$ | outcome variable |
| $Y_i(d)$ | potential outcome for treatment group $d \in \{0,1\}$ |
| $D_i(z)$ | potential treatment choice under assignment $z \in \{0,1\}$ |
| $p(s)$ | proportion of stratum $s \in \mathcal{S}$ |
| $\pi_z(s)$ | treatment assignment probability for treatment group $z \in \{0,1\}$ in stratum $s \in \mathcal{S}$ |
| $n$ | sample size |
| $n_z(s)$ | number of observations in treatment group $z \in \{0,1\}$ in stratum $s$ |
| $n(s)$ | number of observations in stratum $s \in \mathcal{S}$ |
| $\widehat{p}(s)$ | $n(s)/n$, proportion of stratum $s \in \mathcal{S}$ in the sample |
| $\widehat{\pi}_z(s)$ | $n_z(s)/n(s)$, estimated treatment assignment probability for treatment group $z \in \{0,1\}$ in stratum $s \in \mathcal{S}$ |
| $F_{Y(d)}(y)$ | $\mathbb{E}[\mathbb{1}_{\{Y(d) \leq y\}}]$, potential outcome distribution function |
| $\mu_z(y,s,x)$ | $\mathbb{E}[\mathbb{1}_{\{Y \leq y\}} \mid Z = z, S = s, X = x]$, conditional distribution function |
| $\eta_z(s,x)$ | $\mathbb{E}[D \mid Z = z, S = s, X = x]$, conditional probability of treatment receipt |
| $[K]$ | $\{1, \ldots, K\}$ for a positive integer $K$ |
| $\|a\|$ | $\sqrt{a^\top a}$, Euclidean norm of a vector $a = (a_1, \ldots, a_p)^\top \in \mathbb{R}^p$ |
| $\|\cdot\|_{P,q}$ | $L^q(P)$ norm |
| $\ell^\infty(\mathcal{Y})$ | space of uniformly bounded functions mapping an arbitrary index set $\mathcal{Y}$ to the real line |
| $\rightsquigarrow$ | convergence in distribution or law |
| $\overset{d}{=}$ | equality in distribution |
| $X_n = O_p(a_n)$ | $\lim_{K \to \infty} \lim_{n \to \infty} P(|X_n| > Ka_n) = 0$ for a sequence $a_n > 0$ |
| $X_n = o_p(a_n)$ | $\sup_{K > 0} \lim_{n \to \infty} P(|X_n| > Ka_n) = 0$ for a sequence $a_n > 0$ |
| $x_n \lesssim y_n$ | for sequences $x_n$ and $y_n$ in $\mathbb{R}$, $x_n \leq Ay_n$ for a constant $A$ |
| $\lfloor b \rfloor$ | $\max\{k \in \mathbb{Z} \mid k \leq b\}$, greatest integer less than or equal to $b$ |

## B  Definitions

We first introduce some definitions from empirical process theory that will be used in the proofs. See also van der Vaart and Wellner (1996) and Chernozhukov et al. (2014) for more details.

**Definition B.1** (Covering numbers). The *covering number* $N(\varepsilon, \mathcal{F}, \|\cdot\|)$ is the minimal number of balls $\{g : \|g - f\| < \varepsilon\}$ of radius $\varepsilon$ needed to cover the set $\mathcal{F}$. The centers of the balls need not belong to $\mathcal{F}$, but they should have finite norms.

**Definition B.2** (Envelope function). An *envelope function* of a class $\mathcal{F}$ is any function $x \mapsto F(x)$ such that $|f(x)| \leq F(x)$ for every $x$ and $f$.

**Definition B.3** (VC-type class). We say $\mathcal{F}$ is of *VC-type* with coefficients $(\alpha, v)$ and envelope $F$ if the uniform covering numbers satisfy the following:

$$\sup_Q N\left(\varepsilon \|F\|_{Q,2}, \mathcal{F}, L_2(Q)\right) \leq \left(\frac{\alpha}{\varepsilon}\right)^v, \quad \forall \varepsilon \in (0,1],$$

where the supremum is taken over all finitely discrete probability measures.

## C  Proofs

### C.1  Proof of Lemma 3.2

To prove Lemma 3.2, we introduce additional notation to categorize individuals based on their compliance type. Table 3 summarizes the four compliance types with respect to the potential treatment choices. We let $\mathcal{C}$ denote the compliance type, and $\mathcal{C} = c$ denote the compliers, i.e., those with $D(1) > D(0)$.

Table 3: Compliance types

| $D(1)$ | $D(0)$ | type |
|--------|--------|------|
| 0 | 0 | never-takers |
| 0 | 1 | defiers |
| 1 | 0 | compliers |
| 1 | 1 | always-takers |

*Proof.* Under the monotonicity assumption stated in Assumption 3.1(iv), we can identify the cumulative distribution functions of potential outcomes for the compliers conditional on $S$ as follows:

$$F_{Y(1)}(y \mid S, \mathcal{C} = c) = \frac{\mathbb{E}[\mathbb{1}_{\{Y \leq y\}} \cdot D \mid Z = 1, S] - \mathbb{E}[\mathbb{1}_{\{Y \leq y\}} \cdot D \mid Z = 0, S]}{\mathbb{E}[D \mid Z = 1, S] - \mathbb{E}[D \mid Z = 0, S]}, \tag{22}$$

$$F_{Y(0)}(y \mid S, \mathcal{C} = c) = \frac{\mathbb{E}[\mathbb{1}_{\{Y \leq y\}} \cdot (1 - D) \mid Z = 1, S] - \mathbb{E}[\mathbb{1}_{\{Y \leq y\}} \cdot (1 - D) \mid Z = 0, S]}{\mathbb{E}[1 - D \mid Z = 1, S] - \mathbb{E}[1 - D \mid Z = 0, S]}. \tag{23}$$

We can then derive the unconditional CDF of the potential outcomes for the compliers by aggregating over the strata:

$$
\begin{aligned}
F_{Y(1)}(y \mid \mathcal{C} = c) &= \sum_{s=1}^{S} P(S = s \mid \mathcal{C} = c) F_{Y(1)}(y \mid S = s, \mathcal{T} = c) \\
&= \sum_{s=1}^{S} \frac{P(\mathcal{C} = c \mid S = s)}{P(\mathcal{C} = c)} F_{Y(1)}(y \mid S = s, \mathcal{C} = c) \\
&= \frac{\sum_{s=1}^{S} p(s)(\mathbb{E}[\mathbb{1}_{\{Y \leq y\}} \cdot D \mid Z = 1, S = s] - \mathbb{E}[\mathbb{1}_{\{Y \leq y\}} \cdot D \mid Z = 0, S = s])}{\sum_{s=1}^{S} p(s)(\mathbb{E}[D \mid Z = 1, S = s] - \mathbb{E}[D \mid Z = 0, S = s])}.
\end{aligned}
$$

The first equality holds by the law of total expectation. The second equality holds by the Bayes' law. The third equality follows from representation of the conditional distribution given in (22) and the fact that $P(\mathcal{C} = c \mid S = s) = \mathbb{E}[D \mid Z = 1, S = s] - \mathbb{E}[D \mid Z = 0, S = s]$. We can obtain similar expressions for $F_{Y(0)}(y \mid \mathcal{C} = c)$ using the representation given in (23) as follows:

$$F_{Y(0)}(y \mid \mathcal{C} = c) = \frac{\sum_{s=1}^{S} p(s)(\mathbb{E}[\mathbb{1}_{\{Y \leq y\}} \cdot (1 - D) \mid Z = 1, S = s] - \mathbb{E}[\mathbb{1}_{\{Y \leq y\}} \cdot (1 - D) \mid Z = 0, S = s])}{\sum_{s=1}^{S} p(s)(\mathbb{E}[1 - D \mid Z = 1, S = s] - \mathbb{E}[1 - D \mid Z = 0, S = s])}.$$

Then, the LDTE, the difference between the distribution functions is given by

$$
\begin{aligned}
\beta(y) :&= F_{Y(1)}(y \mid \mathcal{C} = c) - F_{Y(0)}(y \mid \mathcal{C} = c) \\
&= \frac{\sum_{s=1}^{S} p(s)(\mathbb{E}[\mathbb{1}_{\{Y \leq y\}} \cdot D \mid Z = 1, S = s] - \mathbb{E}[\mathbb{1}_{\{Y \leq y\}} \cdot D \mid Z = 0, S = s])}{\sum_{s=1}^{S} p(s)(\mathbb{E}[D \mid Z = 1, S = s] - \mathbb{E}[D \mid Z = 0, S = s])} \\
&\quad + \frac{\sum_{s=1}^{S} p(s)(\mathbb{E}[\mathbb{1}_{\{Y \leq y\}} \cdot (1 - D) \mid Z = 1, S = s] - \mathbb{E}[\mathbb{1}_{\{Y \leq y\}} \cdot (1 - D) \mid Z = 0, S = s])}{\sum_{s=1}^{S} p(s)(\mathbb{E}[D \mid Z = 1, S = s] - \mathbb{E}[D \mid Z = 0, S = s])} \\
&= \frac{\sum_{s=1}^{S} p(s)(\mathbb{E}[\mathbb{1}_{\{Y \leq y\}} \mid Z = 1, S = s] - \mathbb{E}[\mathbb{1}_{\{Y \leq y\}} \mid Z = 0, S = s])}{\sum_{s=1}^{S} p(s)(\mathbb{E}[D \mid Z = 1, S = s] - \mathbb{E}[D \mid Z = 0, S = s])}.
\end{aligned}
$$

This completes the proof. $\square$

## C.2 Proof of Theorem 5.2

*Proof.* Let

$$
\begin{aligned}
B &:= \mathbb{E}[D(1) - D(0)], \\
T(y) &:= \mathbb{E}[(\mathbb{1}_{\{Y(1) \leq y\}} - \mathbb{1}_{\{Y(0) \leq y\}})(D(1) - D(0))], \\
\widehat{B} &:= \frac{1}{n} \sum_{i=1}^{n} (\Xi_{1,i}^{D} - \Xi_{0,i}^{D}), \\
\widehat{T}(y) &:= \frac{1}{n} \sum_{i=1}^{n} (\Xi_{1,i}^{Y}(y) - \Xi_{0,i}^{Y}(y)).
\end{aligned}
$$

Then, we have

$$
\begin{aligned}
\sqrt{n}\left(\widehat{\beta}(y) - \beta(y)\right) &= \sqrt{n}\left(\frac{\widehat{T}(y)}{\widehat{B}} - \frac{T(y)}{B}\right) \\
&= \frac{1}{\widehat{B}}\sqrt{n}\left(\widehat{T}(y) - T(y)\right) - \frac{T(y)}{\widehat{B}B}\sqrt{n}\left(\widehat{B} - B\right) \\
&= \frac{1}{\widehat{B}}\left[\sqrt{n}\left(\widehat{T}(y) - T(y)\right) - \beta(y)\sqrt{n}\left(\widehat{B} - B\right)\right]. \quad (24)
\end{aligned}
$$

**Step 1.** First, we start with the linear expansion of $\sqrt{n}\left(\widehat{T}(y) - T(y)\right)$.

$$
\begin{aligned}
\sqrt{n}(\widehat{T}(y) - T(y)) &= \frac{1}{\sqrt{n}} \sum_{i=1}^{n} \left[\frac{Z_i \cdot (\mathbb{1}_{\{Y_i \leq y\}} - \widehat{\mu}_1(y, S_i, X_i))}{\widehat{\pi}_1(S_i)} - \frac{(1 - Z_i) \cdot (\mathbb{1}_{\{Y_i \leq y\}} - \widehat{\mu}_0(y, S_i, X_i))}{\widehat{\pi}_0(S_i)}\right. \\
&\qquad \left. + \widehat{\mu}_1(y, S_i, X_i) - \widehat{\mu}_0(y, S_i, X_i)\right] - \sqrt{n}T(y) \\
&= \underbrace{\frac{1}{\sqrt{n}} \sum_{i=1}^{n} \left[\widehat{\mu}_1(y, S_i, X_i) - \frac{Z_i\widehat{\mu}_1(y, S_i, X_i))}{\widehat{\pi}_1(S_i)}\right]}_{\equiv T_{n,1}} \\
&\quad + \underbrace{\frac{1}{\sqrt{n}} \sum_{i=1}^{n} \left[\frac{(1 - Z_i)\widehat{\mu}_0(y, S_i, X_i)}{\widehat{\pi}_0(S_i)} - \widehat{\mu}_0(y, S_i, X_i)\right]}_{\equiv T_{n,2}} \\
&\quad + \underbrace{\frac{1}{\sqrt{n}} \sum_{i=1}^{n} \frac{Z_i \cdot \mathbb{1}_{\{Y_i \leq y\}}}{\widehat{\pi}_1(S_i)} - \frac{1}{\sqrt{n}} \sum_{i=1}^{n} \frac{(1 - Z_i) \cdot \mathbb{1}_{\{Y_i \leq y\}}}{1 - \widehat{\pi}_1(S_i)} - \sqrt{n}T(y)}_{\equiv T_{n,3}}. \quad (25)
\end{aligned}
$$

We start with the first term $T_{n,1}$ in (25).

$$T_{n,1} = \frac{1}{\sqrt{n}} \sum_{i=1}^{n} \left[ \widehat{\mu}_1(y, S_i, X_i) - \frac{Z_i \widehat{\mu}_1(y, S_i, X_i))}{\widehat{\pi}_1(S_i)} \right]$$

$$= -\frac{1}{\sqrt{n}} \sum_{i=1}^{n} \frac{Z_i - \widehat{\pi}_1(S_i)}{\widehat{\pi}_1(S_i)} \widehat{\mu}_1(y, S_i, X_i)$$

$$= -\frac{1}{\sqrt{n}} \sum_{i=1}^{n} \frac{Z_i - \widehat{\pi}_1(S_i)}{\widehat{\pi}_1(S_i)} \left[ \widehat{\mu}_1(y, S_i, X_i) - \mu_1(y, S_i, X_i) + \mu_1(y, S_i, X_i) \right]$$

$$= -\frac{1}{\sqrt{n}} \sum_{i=1}^{n} \frac{Z_i - \widehat{\pi}_1(S_i)}{\widehat{\pi}_1(S_i)} \delta_1^Y(y, S_i, X_i) - \frac{1}{\sqrt{n}} \sum_{i=1}^{n} \frac{Z_i}{\widehat{\pi}_1(S_i)} \mu_1(y, S_i, X_i) + \frac{1}{\sqrt{n}} \sum_{i=1}^{n} \mu_1(y, S_i, X_i)$$

$$= -\frac{1}{\sqrt{n}} \sum_{i=1}^{n} \frac{Z_i - \widehat{\pi}_1(S_i)}{\widehat{\pi}_1(S_i)} \delta_1^Y(y, S_i, X_i) - \frac{1}{\sqrt{n}} \sum_{i=1}^{n} \frac{Z_i}{\widehat{\pi}_1(S_i)} \tilde{\mu}_1(y, S_i, X_i) + \frac{1}{\sqrt{n}} \sum_{i=1}^{n} \tilde{\mu}_1(y, S_i, X_i)$$

$$= \frac{1}{\sqrt{n}} \sum_{i=1}^{n} \left( 1 - \frac{1}{\pi_1(S_i)} \right) Z_i \tilde{\mu}_1(y, S_i, X_i) + \frac{1}{\sqrt{n}} \sum_{i=1}^{n} (1 - Z_i) \tilde{\mu}_1(y, S_i, X_i)$$

$$+ \underbrace{\frac{1}{\sqrt{n}} \sum_{s \in \mathcal{S}} \left( \frac{\widehat{\pi}_1(s) - \pi_1(s)}{\widehat{\pi}_1(s) \pi_1(s)} \right) \left( \sum_{i=1}^{n} Z_i \tilde{\mu}_1(y, s, X_i) \mathbb{1}\{S_i = s\} \right)}_{\equiv R_{1,1}(y)} - \underbrace{\frac{1}{\sqrt{n}} \sum_{i=1}^{n} \frac{Z_i - \widehat{\pi}_1(S_i)}{\widehat{\pi}_1(S_i)} \delta_1^Y(y, S_i, X_i)}_{\equiv R_{1,2}(y)},$$

where the second last equality holds because we have

$$\frac{1}{\sqrt{n}} \sum_{i=1}^{n} \frac{Z_i}{\widehat{\pi}_1(S_i)} \mathbb{E}[\mu_1(y, S_i, X_i) \mid S_i] = \frac{1}{\sqrt{n}} \sum_{i=1}^{n} \mathbb{E}[\mu_1(y, S_i, X_i) \mid S_i].$$

Let $B_n(s) := \sum_{i=1}^{n} (Z_i - \pi_1(s)) \cdot \mathbb{1}\{S_i = s\}$. Note that we have $\widehat{\pi}_1(s) - \pi_1(s) = \frac{B_n(s)}{n(s)}$. For the first term $R_{1,1}(y)$, we have

$$\sup_{y \in \mathcal{Y}} \left| \frac{1}{\sqrt{n}} \sum_{s \in \mathcal{S}} \left( \frac{\pi_1(s) - \hat{\pi}_1(s)}{\hat{\pi}_1(s) \pi_1(s)} \right) \left( \sum_{i=1}^{n} Z_i \tilde{\mu}_1(y, s, X_i) \mathbb{1}\{S_i = s\} \right) \right|$$

$$\leq \sum_{s \in \mathcal{S}} \left| \frac{B_n(s)}{n_1(s) \pi_1(s)} \right| \sup_{y \in \mathcal{Y}, s \in \mathcal{S}} \left| \frac{1}{\sqrt{n}} \sum_{i=1}^{n} Z_i \tilde{\mu}_1(y, s, X_i) \mathbb{1}\{S_i = s\} \right|.$$

Assumption 5.1 implies that the class $\{\tilde{\mu}_1(y, s, X_i) : y \in \mathcal{Y}\}$ is of the VC-type with fixed coefficients $(\alpha, v)$ and an envelope $F_i$ such that $\mathbb{E}(|F_i|^d | S_i = s) < \infty$ for $d > 2$. Therefore,

$$\sup_{y \in \mathcal{Y}, s \in \mathcal{S}} \left| \frac{1}{\sqrt{n}} \sum_{i=1}^{n} Z_i \tilde{\mu}_1(y, s, X_i) \mathbb{1}\{S_i = s\} \right| = O_p(1).$$

It is also assumed that $\widehat{\pi}_1(s) - \pi_1(s) = o_p(1)$ and $n(s)/n_1(s) \xrightarrow{p} 1/\pi_1(s) < \infty$. Therefore, we have

$$\sup_{y \in \mathcal{Y}} |R_{1,1}(y)| = o_p(1).$$

Now, consider the term $R_{1,2}(y)$:

$$
\left| \frac{1}{\sqrt{n}} \sum_{i=1}^{n} \frac{Z_i - \widehat{\pi}_1(S_i)}{\widehat{\pi}_1(S_i)} \delta_1^Y(y, S_i, X_i) \right| = \left| \frac{1}{\sqrt{n}} \sum_{s \in \mathcal{S}} \sum_{i=1}^{n} \frac{Z_i - \widehat{\pi}_1(s)}{\widehat{\pi}_1(s)} \delta_1^Y(y, s, X_i) \mathbb{1}\{S_i = s\} \right|
$$

$$
= \frac{1}{\sqrt{n}} \left| \sum_{s \in \mathcal{S}} \frac{1}{\widehat{\pi}_1(s)} \sum_{i=1}^{n} Z_i \delta_1^Y(y, s, X_i) \mathbb{1}\{S_i = s\} - \sum_{s \in \mathcal{S}} \sum_{i=1}^{n} \delta_1^Y(y, s, X_i) \mathbb{1}\{S_i = s\} \right|
$$

$$
= \frac{1}{\sqrt{n}} \left| \sum_{s \in \mathcal{S}} \sum_{i \in I_1(s)} \delta_1^Y(y, s, X_i) \frac{n(s)}{n_1(s)} - \sum_{s \in \mathcal{S}} \sum_{i \in I_0(s) \cup I_1(s)} \delta_1^Y(y, s, X_i) \right|
$$

$$
= \frac{1}{\sqrt{n}} \left| \sum_{s \in \mathcal{S}} \sum_{i \in I_1(s)} \delta_1^Y(y, s, X_i) \frac{n_0(s)}{n_1(s)} - \sum_{s \in \mathcal{S}} \sum_{i \in I_0(s)} \delta_1^Y(y, s, X_i) \right|
$$

$$
= \frac{1}{\sqrt{n}} \left| \sum_{s \in \mathcal{S}} n_0(s) \left[ \frac{\sum_{i \in I_1(s)} \delta_1^Y(y, s, X_i)}{n_1(s)} - \frac{\sum_{i \in I_0(s)} \delta_1^Y(y, s, X_i)}{n_0(s)} \right] \right|
$$

$$
\leq \frac{1}{\sqrt{n}} \sum_{s \in \mathcal{S}} n_0(s) \sup_{y \in \mathcal{Y}} \left| \frac{\sum_{i \in I_1(s)} \delta_1^Y(y, s, X_i)}{n_1(s)} - \frac{\sum_{i \in I_0(s)} \delta_1^Y(y, s, X_i)}{n_0(s)} \right| = o_p(1)
$$

where the last equality is due to Assumption 5.1 (i).

Therefore, we have

$$
T_{n,1} = \frac{1}{\sqrt{n}} \sum_{i=1}^{n} \left( 1 - \frac{1}{\pi_1(S_i)} \right) Z_i \tilde{\mu}_1(y, S_i, X_i) + \frac{1}{\sqrt{n}} \sum_{i=1}^{n} (1 - Z_i) \tilde{\mu}_1(y, S_i, X_i) + R_1(y),
$$

where $\sup_{y \in \mathcal{Y}} R_1(y) = o_p(1)$.

The linear expansion of $T_{n,2}$ can be established in the same manner. As for the third term $T_{n,3}$, first note that

$$
\frac{1}{\sqrt{n}} \sum_{i=1}^{n} \frac{\mathbb{1}_{\{Z_i = z\}} \cdot \mathbb{1}_{\{Y_i \leq y\}}}{\widehat{\pi}_z(S_i)} = \frac{1}{\sqrt{n}} \sum_{i=1}^{n} \frac{\mathbb{1}_{\{Z_i = z\}} \cdot \mathbb{1}_{\{Y_i(D_i(z)) \leq y\}}}{\widehat{\pi}_z(S_i)} =: \frac{1}{\sqrt{n}} \sum_{i=1}^{n} \frac{\mathbb{1}_{\{Z_i = z\}} \cdot Y_i^z(y)}{\widehat{\pi}_z(S_i)}.
$$

Then we have

$$
T_{n,3} = \frac{1}{\sqrt{n}} \sum_{i=1}^{n} \frac{Z_i \cdot \mathbb{1}_{\{Y_i \leq y\}}}{\widehat{\pi}_1(S_i)} - \frac{1}{\sqrt{n}} \sum_{i=1}^{n} \frac{(1 - Z_i) \cdot \mathbb{1}_{\{Y_i \leq y\}}}{\widehat{\pi}_0(S_i)} - \sqrt{n} T(y)
$$

$$
= \left\{ \frac{1}{\sqrt{n}} \sum_{i=1}^{n} \frac{1}{\widehat{\pi}_1(S_i)} \tilde{Y}_i^1(y) Z_i - \frac{1}{\sqrt{n}} \sum_{i=1}^{n} \frac{1 - Z_i}{\widehat{\pi}_0(S_i)} \tilde{Y}_i^0(y) \right\}
$$

$$
+ \left\{ \frac{1}{\sqrt{n}} \sum_{i=1}^{n} \frac{1}{\widehat{\pi}_1(S_i)} \mathbb{E}[Y_i^1(y)|S_i] Z_i - \frac{1}{\sqrt{n}} \sum_{i=1}^{n} \frac{1 - Z_i}{\widehat{\pi}_0(S_i)} \mathbb{E}[Y_i^0(y)|S_i] - \sqrt{n} T(y) \right\}. \quad (26)
$$

First note that

$$
\frac{1}{\sqrt{n}} \sum_{i=1}^{n} \frac{1}{\widehat{\pi}_1(S_i)} \mathbb{E}[Y_i^1(y)|S_i] Z_i = \frac{1}{\sqrt{n}} \sum_{i=1}^{n} \frac{1}{\pi_1(S_i)} \mathbb{E}[Y_i^1(y)|S_i] Z_i - \frac{1}{\sqrt{n}} \sum_{i=1}^{n} \frac{\widehat{\pi}_1(S_i) - \pi_1(S_i)}{\widehat{\pi}_1(S_i)\pi_1(S_i)} \mathbb{E}[Y_i^1(y)|S_i] Z_i,
$$

$$\frac{1}{\sqrt{n}} \sum_{i=1}^{n} \frac{1}{\pi_1(S_i)} \mathbb{E}[Y_i^1(y)|S_i] Z_i = \sum_{s \in \mathcal{S}} \frac{1}{\sqrt{n}} \sum_{i=1}^{n} \frac{1}{\pi_1(s)} \mathbb{E}[Y_i^1(y)|S_i = s] Z_i 1\{S_i = s\}$$

$$= \sum_{s \in \mathcal{S}} \frac{1}{\sqrt{n}} \sum_{i=1}^{n} \frac{\mathbb{E}[Y_i^1(y)|S_i = s]}{\pi_1(s)} (Z_i - \pi_1(s)) 1\{S_i = s\}$$

$$+ \sum_{s \in \mathcal{S}} \frac{1}{\sqrt{n}} \sum_{i=1}^{n} \frac{1}{\pi_1(s)} \mathbb{E}[Y_i^1(y)|S_i = s] \pi_1(s) 1\{S_i = s\}$$

$$= \sum_{s \in \mathcal{S}} \frac{\mathbb{E}[Y^1(y)|S = s]}{\pi_1(s)\sqrt{n}} \sum_{i=1}^{n} (Z_i - \pi_1(s)) 1\{S_i = s\}$$

$$+ \sum_{s \in \mathcal{S}} \frac{\mathbb{E}[Y^1(y)|S = s]}{\sqrt{n}} \sum_{i=1}^{n} 1\{S_i = s\}$$

$$= \sum_{s \in \mathcal{S}} \frac{\mathbb{E}[Y^1(y)|S = s]}{\pi_1(s)\sqrt{n}} B_n(s) + \sum_{s \in \mathcal{S}} \frac{\mathbb{E}[Y^1(y)|S = s]}{\sqrt{n}} n(s),$$

and

$$\frac{1}{\sqrt{n}} \sum_{i=1}^{n} \frac{\widehat{\pi}_1(S_i) - \pi_1(S_i)}{\widehat{\pi}_1(S_i)\pi_1(S_i)} \mathbb{E}[Y_i^1(y)|S_i] Z_i = \sum_{s \in \mathcal{S}} \frac{1}{\sqrt{n}} \sum_{i=1}^{n} \frac{\widehat{\pi}(s) - \pi_1(s)}{\widehat{\pi}(s)\pi_1(s)} \mathbb{E}[Y_i^1(y)|S_i = s] Z_i 1\{S_i = s\}$$

$$= \sum_{s \in \mathcal{S}} \frac{1}{\sqrt{n}} \sum_{i=1}^{n} \frac{B_n(s)}{n(s)\widehat{\pi}(s)\pi_1(s)} \mathbb{E}[Y_i^1(y)|S_i = s] Z_i 1\{S_i = s\}$$

$$= \sum_{s \in \mathcal{S}} \frac{B_n(s)\mathbb{E}[Y^1(y)|S = s]}{\sqrt{n} n(s)\widehat{\pi}(s)\pi_1(s)} \sum_{i=1}^{n} Z_i 1\{S_i = s\}$$

$$= \sum_{s \in \mathcal{S}} \frac{B_n(s)\mathbb{E}[Y^1(y)|S = s]}{\sqrt{n} n(s)\widehat{\pi}(s)\pi_1(s)} n_1(s)$$

$$= \sum_{s \in \mathcal{S}} \frac{B_n(s)\mathbb{E}[Y^1(y)|S = s]}{\sqrt{n}\pi_1(s)}.$$

Therefore, we have

$$\frac{1}{\sqrt{n}} \sum_{i=1}^{n} \frac{1}{\widehat{\pi}_1(S_i)} \mathbb{E}[Y_i^1(y)|S_i] Z_i = \sum_{s \in \mathcal{S}} \frac{\mathbb{E}[Y^1(y)|S = s]}{\sqrt{n}} n(s).$$

Similarly, we have

$$\frac{1}{\sqrt{n}} \sum_{i=1}^{n} \frac{1 - Z_i}{\widehat{\pi}_0(S_i)} \mathbb{E}[Y_i^0(y)|S_i] = \sum_{s \in \mathcal{S}} \frac{\mathbb{E}[Y^0(y)|S = s]}{\sqrt{n}} n(s)$$

Then, we have

$$\frac{1}{\sqrt{n}} \sum_{i=1}^{n} \frac{1}{\widehat{\pi}_1(S_i)} \mathbb{E}[Y_i^1(y)|S_i] Z_i - \frac{1}{\sqrt{n}} \sum_{i=1}^{n} \frac{1 - Z_i}{1 - \widehat{\pi}_1(S_i)} \mathbb{E}[Y_i^0(y)|S_i] - \sqrt{n} T(y)$$

$$= \sum_{s \in \mathcal{S}} \frac{\mathbb{E}[Y^1(y)|S = s]}{\sqrt{n}} n(s) - \sum_{s \in \mathcal{S}} \frac{\mathbb{E}[Y^0(y)|S = s]}{\sqrt{n}} n(s) - \sqrt{n} T(y)$$

$$= \sum_{s \in \mathcal{S}} \sqrt{n} \left( \frac{n(s)}{n} - p(s) \right) \mathbb{E}[Y^1(y) - Y^0(y)|S = s] + \sum_{s \in \mathcal{S}} \sqrt{n} p(s) \mathbb{E}[Y^1(y) - Y^0(y)|S = s] - \sqrt{n} T(y)$$

$$= \sum_{s \in \mathcal{S}} \sqrt{n} \left( \frac{n(s)}{n} - p(s) \right) \mathbb{E}[Y^1(y) - Y^0(y)|S = s] + \sqrt{n} \mathbb{E}[Y^1(y) - Y^0(y)] - \sqrt{n} T(y)$$

$$= \sum_{s \in \mathcal{S}} \frac{n(s)}{\sqrt{n}} \mathbb{E}[Y^1(y) - Y^0(y)|S = s] - \sqrt{n} \mathbb{E}[Y^1(y) - Y^0(y)]$$

$$= \frac{1}{\sqrt{n}} \sum_{s \in \mathcal{S}} \sum_{i=1}^{n} \left( 1\{S_i = s\} \mathbb{E}[Y_i^1(y) - Y_i^0(y)|S_i = s] \right) - \sqrt{n} \mathbb{E}[Y^1(y) - Y^0(y)]$$

$$= \frac{1}{\sqrt{n}} \sum_{i=1}^{n} \mathbb{E}[Y_i^1(y) - Y_i^0(y)|S_i] - \sqrt{n} \mathbb{E}[Y^1(y) - Y^0(y)]$$

$$= \frac{1}{\sqrt{n}} \sum_{i=1}^{n} \left( \mathbb{E}[Y_i^1(y) - Y_i^0(y)|S_i] - \mathbb{E}[Y_i^1(y) - Y_i^0(y)] \right). \tag{27}$$

Combining, we have

$$T_{n,3} = \left\{ \frac{1}{\sqrt{n}} \sum_{i=1}^{n} \frac{1}{\widehat{\pi}_1(S_i)} \tilde{Y}_i^1(y) Z_i - \frac{1}{\sqrt{n}} \sum_{i=1}^{n} \frac{1 - Z_i}{1 - \widehat{\pi}_1(S_i)} \tilde{Y}_i^0(y) \right\}$$

$$+ \left\{ \frac{1}{\sqrt{n}} \sum_{i=1}^{n} \left( \mathbb{E}[Y_i^1(y) - Y_i^0(y)|S_i] - \mathbb{E}[Y_i^1(y) - Y_i^0(y)] \right) \right\}$$

$$= \left\{ \frac{1}{\sqrt{n}} \sum_{i=1}^{n} \frac{1}{\pi_1(S_i)} \tilde{Y}_i^1(y) Z_i - \frac{1}{\sqrt{n}} \sum_{i=1}^{n} \frac{1 - Z_i}{\pi_0(S_i)} \tilde{Y}_i^0(y) \right\}$$

$$+ \left\{ \frac{1}{\sqrt{n}} \sum_{i=1}^{n} \left( \mathbb{E}[Y_i^1(y) - Y_i^0(y)|S_i] - \mathbb{E}[Y_i^1(y) - Y_i^0(y)] \right) \right\} + R(3),$$

where $\sup_{y \in \mathcal{Y}} R_3(y) = o_p(1)$. This is because we have for $z \in \{0, 1\}$,

$$\sup_{y \in \mathcal{Y}, s \in \mathcal{S}} \left| \left( \frac{1}{\pi_z(s)} - \frac{1}{\widehat{\pi}_z(s)} \right) \frac{1}{\sqrt{n}} \sum_{i=1}^{n} \tilde{Y}_i^z(y) \mathbb{1}\{Z_i = z\} \mathbb{1}\{S_i = s\} \right| = o_p(1)$$

due to the same argument used in the proofs of $T_{n,1}$.

Hence, combining we have

$$\sqrt{n}(\widehat{T}(y) - T(y)) = \left\{ \frac{1}{\sqrt{n}} \sum_{i=1}^{n} \left[ \left( 1 - \frac{1}{\pi_1(S_i)} \right) \tilde{\mu}_1(y, S_i, X_i) - \tilde{\mu}_0(y, S_i, X_i) + \frac{\tilde{Y}_i^1(y)}{\pi_1(S_i)} \right] Z_i \right.$$

$$+ \frac{1}{\sqrt{n}} \sum_{i=1}^{n} \left[ \left( \frac{1}{\pi_0(S_i)} - 1 \right) \tilde{\mu}_0(y, S_i, X_i) + \tilde{\mu}_1(y, S_i, X_i) - \frac{\tilde{Y}_i^0}{\pi_0(S_i)} \right] (1 - Z_i) \right\}$$

$$+ \left\{ \frac{1}{\sqrt{n}} \sum_{i=1}^{n} \left( \mathbb{E}[Y_i^1(y) - Y_i^0(y)|S_i] - \mathbb{E}[Y_i^1(y) - Y_i^0(y)] \right) \right\} + R(y), \tag{28}$$

where $\sup_{y \in \mathcal{Y}} |R(y)| = o_p(1)$.

**Step 2.** Using the same arguments, we can show that

$$\sqrt{n}(\widehat{B} - B) = \left\{ \frac{1}{\sqrt{n}} \sum_{i=1}^{n} \left[ \left( 1 - \frac{1}{\pi_1(S_i)} \right) \tilde{\eta}_1(S_i, X_i) - \tilde{\eta}_0(S_i, X_i) + \frac{\tilde{D}_i(1)}{\pi_1(S_i)} \right] Z_i \right.$$

$$+ \frac{1}{\sqrt{n}} \sum_{i=1}^{n} \left[ \left( \frac{1}{\pi_0(S_i)} - 1 \right) \tilde{\eta}_0(S_i, X_i) + \tilde{\eta}_1(S_i, X_i) - \frac{\tilde{D}_i(0)}{\pi_0(S_i)} \right] (1 - Z_i) \right\}$$

$$+ \left\{ \frac{1}{\sqrt{n}} \sum_{i=1}^{n} \left( \mathbb{E}[D_i(1) - D_i(0)|S_i] - \mathbb{E}[D_i(1) - D_i(0)] \right) \right\} + o_p(1). \qquad (29)$$

**Step 3.** Let $\mathcal{D}_i := \{Y_i(1), Y_i(0), D_i(1), D_i(0), X_i\}$. Define, for $z \in \{0, 1\}$,

$$\phi_z(y, S_i, \mathcal{D}_i) := \left( 1 - \frac{1}{\pi_z(S_i)} \right) \tilde{\mu}_z(y, S_i, X_i) - \tilde{\mu}_{1-z}(y, S_i, X_i) + \frac{\tilde{Y}_i^z(y)}{\pi_z(S_i)}$$

$$- \beta(y) \left( \left( 1 - \frac{1}{\pi_z(S_i)} \right) \tilde{\eta}_z(S_i, X_i) - \tilde{\eta}_{1-z}(S_i, X_i) + \frac{\tilde{D}_i(z)}{\pi_z(S_i)} \right), \qquad (30)$$

and

$$\xi_i(y) := \mathbb{E}[Y_i^1(y) - Y_i^0(y)|S_i] - \mathbb{E}[Y_i^1(y) - Y_i^0(y)]$$

$$- \beta(y) \left( \mathbb{E}[D_i(1) - D_i(0)|S_i] - \mathbb{E}[D_i(1) - D_i(0)] \right). \qquad (31)$$

Combining (28) and (29) into (24), we obtain the linear expansion for $\widehat{\beta}(y)$ as

$$\sqrt{n} \left( \widehat{\beta}(y) - \beta(y) \right)$$

$$= \frac{1}{\widehat{B}} \left[ \sqrt{n} \left( \widehat{T}(y) - T(y) \right) - \beta(y) \sqrt{n} \left( \widehat{B} - B \right) \right]$$

$$= \frac{1}{\widehat{B}} \left[ \frac{1}{\sqrt{n}} \sum_{i=1}^{n} \phi_1(y, S_i, \mathcal{D}_i) Z_i - \frac{1}{\sqrt{n}} \sum_{i=1}^{n} \phi_0(y, S_i, \mathcal{D}_i)(1 - Z_i) + \frac{1}{\sqrt{n}} \sum_{i=1}^{n} \xi_i(y) \right] + I(y)$$

where $\sup_{y \in \mathcal{Y}} |I(y)| = o_p(1)$.

**Step 4.** Denote

$$\varphi_{n,1}(y) := \frac{1}{\sqrt{n}} \sum_{i=1}^{n} \phi_1(y, S_i, \mathcal{D}_i) Z_i - \frac{1}{\sqrt{n}} \sum_{i=1}^{n} \phi_0(y, S_i, \mathcal{D}_i)(1 - Z_i),$$

$$\varphi_{n,2}(y) := \frac{1}{\sqrt{n}} \sum_{i=1}^{n} \xi_i(y)$$

Uniformly over $y \in \mathcal{Y}$, we show that

$$(\varphi_{n,1}(y), \varphi_{n,2}(y)) \rightsquigarrow (\mathcal{G}_1(y), \mathcal{G}_2(y)),$$

where $(\mathcal{G}_1(y), \mathcal{G}_2(y))$ are two independent Gaussian processes with covariance kernels $\Omega_0(y, y') + \Omega_1(y, y')$ and $\Omega_2(y, y')$, respectively, such that

$$\Omega_z(y, y') = \mathbb{E}[\pi_z(S_i) \phi_z(y, S_i, \mathcal{D}_i) \phi_z(y', S_i, \mathcal{D}_i)], z \in \{0, 1\},$$

$$\Omega_2(y, y') = \mathbb{E}[\xi_i(y) \xi_i(y')].$$

The following argument follows the argument provided in the proof of Bugni et al. (2018, Lemma B.2). Note that under Assumption 3.1 (i), conditional on $\{Z_i, S_i\}_{i=1}^{n}$, the distribution of $\varphi_{n,1}(y)$ is the same as the distribution of the same quantity with units ordered by strata $s \in \mathcal{S}$ and then ordered by $Z_i = 1$ first and $Z_i = 0$ second within strata. Let $\{\mathcal{D}_i^s\}_{i=1}^{n}$ be a sequence of i.i.d. random variables with marginal distributions equal to the distribution of $\mathcal{D}_i|S_i = s$. Then we have

$$\varphi_{n,1}(y)|\{Z_i, S_i\}_{i=1}^{n} \overset{d}{=} \widetilde{\varphi}_{n,1}(y)|\{Z_i, S_i\}_{i=1}^{n}$$

where

$$\widetilde{\varphi}_{n,1}(y) := \sum_{s \in \mathcal{S}} \frac{1}{\sqrt{n}} \sum_{i=N(s)+1}^{N(s)+n_1(s)} \phi_1(y, s, \mathcal{D}_i^s) - \sum_{s \in \mathcal{S}} \frac{1}{\sqrt{n}} \sum_{i=N(s)+n_1(s)+1}^{N(s)+n(s)} \phi_0(y, s, \mathcal{D}_i^s).$$

As $\varphi_{n,2}(y)$ is a function of $\{Z_i, S_i\}_{i=1}^n$, we have

$$(\varphi_{n,1}(y), \varphi_{n,2}(y)) \stackrel{d}{=} (\widetilde{\varphi}_{n,1}(y), \varphi_{n,2}(y)).$$

Next, define

$$\varphi_{n,1}^\star(y) := \sum_{s \in \mathcal{S}} \frac{1}{\sqrt{n}} \sum_{i=\lfloor nF(s)\rfloor+1}^{\lfloor n(F(s)+\pi_1(s)p(s))\rfloor} \phi_1(y, s, \mathcal{D}_i^s) - \sum_{s \in \mathcal{S}} \frac{1}{\sqrt{n}} \sum_{i=\lfloor n(F(s)+\pi_1(s)p(s))\rfloor+1}^{\lfloor n(F(s)+p(s))\rfloor} \phi_0(y, s, \mathcal{D}_i^s).$$

Note $\varphi_{n,1}^\star(y)$ is a function of $\{\mathcal{D}_i^s\}_{i\in[n],s\in\mathcal{S}}$, which is independent of $\{Z_i, S_i\}_{i=1}^n$ by construction. Therefore,

$$\varphi_{n,1}^\star(y) \perp\!\!\!\perp \varphi_{n,2}(y).$$

Note that

$$\frac{N(s)}{n} \xrightarrow{p} F(s), \quad \frac{n_1(s)}{n} \xrightarrow{p} \pi_1(s)p(s), \quad \text{and} \quad \frac{n(s)}{n} \xrightarrow{p} p(s).$$

We shall show that

$$\sup_{y\in\mathcal{Y}} |\widetilde{\varphi}_{n,1}(y) - \varphi_{n,1}^\star(y)| = o_p(1) \text{ and } \varphi_{n,1}^\star(y) \rightsquigarrow \mathcal{G}_1(y).$$

We fix $(s, z) \in \mathcal{S} \times \{0, 1\}$ in the remainder of the proof. Define

$$\Gamma_n(s, t, \phi_z) := \frac{1}{\sqrt{n}} \sum_{i=1}^n \mathbb{1}\{i \le \lfloor nt \rfloor\} \cdot \phi_z(y, s, \mathcal{D}_i^s),$$

for $t \in (0, 1]$. The function $\phi_z(y, s, \mathcal{D}_i^s)$ defined in equation (30) can be decomposed as a weighted sum of bounded random functions indexed by $y \in \mathcal{Y}$ with bounded weight functions. More precisely, the class $\mathcal{F} := \{\phi_z(y, s, \mathcal{D}_i^s) : y \in \mathcal{Y}\}$ consists of functions from the following function classes: $\mathcal{F}_1 := \{y \mapsto \tilde{Y}_i^z(y)\}$ and $\mathcal{F}_2 := \{y \mapsto \tilde{\mu}_z(y, s, X_i)\}$. We can show that the class $\mathcal{F}_1$ is Donsker, for instance, by using the bounded, monotone property as established in Theorem 2.7.5 of van der Vaart and Wellner (1996). Also, under Assumption 5.1(ii), Theorem 2.5.2 of van der Vaart and Wellner (1996) yields that $\mathcal{F}_2$ is Donsker. Since all the random weights are uniformly bounded, Corollary 2.10.13 of van der Vaart and Wellner (1996) shows that $\mathcal{F}$ is Donsker. Also, the class $\{t \mapsto \mathbb{1}\{i \le \lfloor nt \rfloor\}$ is VC class and hence Donsker. Since Theorem 2.10.6 of van der Vaart and Wellner (1996) shows that products of uniformly bounded Donsker classes are Donsker, we conclude that the indexed process $\{\Gamma_n(s, t, \phi_z) : t \in (0, 1], \phi_z \in \mathcal{F}\}$ is Donsker. Hence, the result follows.

Next, for a given $y$, by the triangular array central limit theorem,

$$\varphi_{n,1}^\star(y) \rightsquigarrow N(0, \Omega_0(y, y) + \Omega_1(y, y)),$$

where

$$\begin{aligned}
\Omega_0(y, y) + \Omega_1(y, y) &= \lim_{n\to\infty} \sum_{s\in\mathcal{S}} \frac{(\lfloor n(F(s) + \pi_1(s)p(s))\rfloor - \lfloor nF(s)\rfloor)}{n} \mathbb{E}[\phi_1^2(y, s, \mathcal{D}_i^s)] \\
&\quad + \lim_{n\to\infty} \sum_{s\in\mathcal{S}} \frac{(\lfloor n(F(s) + p(s))\rfloor - \lfloor n(F(s) + p(s)\pi_1(s))\rfloor)}{n} \mathbb{E}[\phi_0^2(y, s, \mathcal{D}_i^s)] \\
&= \sum_{s\in\mathcal{S}} p(s)\mathbb{E}[\pi_1(s)\phi_1^2(y, S_i, \mathcal{D}_i) + \pi_0(s)\phi_0^2(y, S_i, \mathcal{D}_i)|S_i = s] \\
&= \mathbb{E}[\pi_1(S_i)\phi_1^2(y, S_i, \mathcal{D}_i)] + \mathbb{E}[\pi_0(S_i)\phi_0^2(y, S_i, \mathcal{D}_i)].
\end{aligned}$$

The finite dimensional convergence follows from the Cramér-Wold device. In particular, the covariance kernel is given by

$$\Omega_0(y, y') + \Omega_1(y, y') = \mathbb{E}[\pi_1(S_i)\phi_1(y, S_i, \mathcal{D}_i)\phi_1(y', S_i, \mathcal{D}_i)] + \mathbb{E}[\pi_0(S_i)\phi_0(y, S_i, \mathcal{D}_i)\phi_0(y', S_i, \mathcal{D}_i)].$$

This concludes the proof of finite-dimensional convergence of $\varphi_{n,1}^\star(y)$.

Finally, since $\{\mu_z(y, s, x)(y) : y \in \mathcal{Y}\}$ is of the VC-type with fixed coefficients $(\alpha, v)$ and a constant envelope function, $\{\xi_i(y) : y \in \mathcal{Y}\}$ is a Donsker class and we have

$$\varphi_{n,2}(y) \rightsquigarrow \mathcal{G}_2(y),$$

where $\mathcal{G}_2(y)$ is a Gaussian process with covariance kernel $\Omega_2(y, y') = \mathbb{E}[\xi_i(y)\xi_i(y')]$. This completes the proof of Step 4.

**Step 5.**  Therefore, uniformly over $y \in \mathcal{Y}$, we have

$$\sqrt{n}\left(\widehat{\beta}(y) - \beta(y)\right) \rightsquigarrow \mathcal{G}(y),$$

where $\mathcal{G}(y)$ is a Gaussian process with covariance kernel

$$\Omega(y, y') = \left\{\mathbb{E}[\pi_1(S_i)\phi_1(y, S_i, \mathcal{D}_i,)\phi_1(y', S_i, \mathcal{D}_i)] + \mathbb{E}[\pi_0(S_i)\phi_0(y, S_i, \mathcal{D}_i)\phi_0(y', S_i, \mathcal{D}_i)]\right.$$
$$\left. + \mathbb{E}[\xi_i(y)\xi_i(y')]\right\} / \left\{\mathbb{E}[D(1) - D(0)]^2\right\}.$$

$\square$

### C.3  Proof of Theorem 5.3: Semiparametric Efficiency Bound

*Proof.* Part (a). We follow the approach used in Hahn (1998) and calculate the semiparametric efficiency bound of the LDTE, $\beta(y)$ for a given $y \in \mathcal{Y}$. First, we characterize the tangent space. To that end, the joint density of the observed variables $(Y, D, Z, X, S)$ can be written as:

$$f(y, d, z, x, s) = f(y \mid d, z, x, s)f(d \mid z, x, s)f(z \mid x, s)f(x \mid s)f(s)$$
$$= f(y \mid d, z, x, s)\{\eta_z(x, s)^d \cdot (1 - \eta_z(x, s))^{1-d}\}\{\pi_1(s)^z \cdot (\pi_0(s))^{1-z}\}f(x \mid s)f(s),$$

where $\eta_z(x, s) := P(D = 1 | Z = z, X = x, S = s)$ and $\pi_1(s) = P(Z = 1 | X = x, S = s)$ for all $x \in \mathcal{X}$.

Consider a regular parametric submodel indexed by $\theta$:

$$f(y, d, z, x, s; \theta) = f^{11}(y \mid x, s; \theta)^{dz} f^{10}(y \mid x, s; \theta)^{d(1-z)} f^{01}(y \mid x, s; \theta)^{(1-d)z} f^{00}(y \mid x, s; \theta)^{(1-d)(1-z)}$$
$$\{\eta_z(x, s; \theta)^d \cdot (1 - \eta_z(x, s; \theta))^{1-d}\}\{\pi_1(s; \theta)^z \cdot (\pi_0(s; \theta))^{1-z}\}f(x \mid s; \theta)f(s; \theta),$$

where $f^{dz}(y \mid x, s; \theta) := f(y \mid d, z, x, s; \theta)$. When the parameter takes the true value, $\theta = \theta_0$, $f(y, d, z, x, s; \theta_0) = f(y, d, z, x, s)$.

The corresponding score of $f(y, d, z, x, s; \theta)$ is given by

$$s(y, d, z, x, s; \theta) := \frac{\partial \ln f(y, d, z, x, s; \theta)}{\partial \theta}$$
$$= dz\dot{f}^{11}(y \mid x, s; \theta) + d(1 - z)\dot{f}^{10}(y \mid x, s; \theta)$$
$$+ (1 - d)z\dot{f}^{01}(y \mid x, s; \theta) + (1 - d)(1 - z)\dot{f}^{00}(y \mid x, s; \theta)$$
$$+ \frac{d - \eta_z(x, s; \theta)}{1 - \eta_z(x, s; \theta)}\dot{\eta}_z(x, s; \theta) + \frac{z - \pi_1(s; \theta)}{\pi_0(s; \theta)}\dot{\pi}(s; \theta) + \dot{f}(x, s; \theta) + \dot{f}(s; \theta),$$

where $\dot{f}$ denotes a derivative of the log, i.e, $\dot{f}(x; \theta) = \frac{\partial \ln f(x; \theta)}{\partial \theta}$.

At the true value, the expectation of the score equals zero. The tangent space of the model is the set of functions that are mean zero and satisfy the additive structure of the score:

$$T = \left\{\begin{array}{l} dza^{11}(y \mid x, s) + d(1 - z)a^{10}(y \mid x, s) \\ + (1 - d)za^{01}(y \mid x, s) + (1 - d)(1 - z)a^{00}(y \mid x, s) \\ + (d - \eta_z(x, s))a_\eta(x, z, s) + (z - \pi_1(s))a_\pi(s) + a_x(x, s) + a_s(s) \end{array}\right\}, \tag{32}$$

where $a^{dz}(y|x, s)$, $a_x(x, s)$ and $a_s(s)$ are mean-zero functions and $a_\eta(x, z, s)$ and $a_{\pi 1}(s)$ are square-integrable functions.

The semiparametric variance bound of $\beta(y)$ is given by the variance of the projection of a function $\psi(Y, D, Z, X, S)$ onto the tangent space $T$. This function must have mean zero, finite second order moment and satisfy the following condition for all regular parametric submodels:

$$\frac{\partial \beta(y; F_\theta)}{\partial \theta}\Big|_{\theta=\theta_0} = \mathbb{E}[\psi(Y, D, Z, X, S) \cdot s(Y, D, Z, X, S)]\Big|_{\theta=\theta_0}. \tag{33}$$

If $\psi$ itself already lies in the tangent space, the variance bound is given by $\mathbb{E}[\psi^2]$.

Now, the LDTE is

$$\beta(y) = F_{Y(1)|\mathcal{C}=c}(y) - F_{Y(0)|\mathcal{C}=c}(y).$$

Following Lemma 3.2, it follows that

$$F_{Y(1)|\mathcal{C}=c}(y) = \left\{ \iint (F_{Y|D=1,Z=1,X=x,S=s}(y) \cdot \eta_1(x, s) - F_{Y|D=1,Z=0,X=x,S=s}(y) \cdot \eta_0(x, s)) f(x|s) f(s) dx ds \right\} / P_C$$

$$F_{Y(0)|\mathcal{C}=c}(y) = -\left\{ \iint (F_{Y|D=0,Z=1,X=x,S=s}(y) \cdot \eta_1(x, s) - F_{Y|D=0,Z=0,X=x,S=s}(y) \cdot \eta_0(x, s)) f(x|s) f(s) dx ds \right\} / P_C$$

where $P_C = \iint (\eta_1(x, s) - \eta_0(x, s)) f(x|s) f(s) dx ds$.

We first need to calculate the derivative evaluated at true $\theta_0$:

$$\frac{\partial \beta(y; F_\theta)}{\partial \theta}\Big|_{\theta=\theta_0} = \frac{\partial}{\partial \theta} F_{Y(1)|\mathcal{C}=c}(y; \theta_0) - \frac{\partial}{\partial \theta} F_{Y(0)|\mathcal{C}=c}(y; \theta_0).$$

We have,

$$\frac{\partial}{\partial \theta} F_{Y(1)|\mathcal{C}=c}(y; \theta_0)$$
$$= \frac{1}{P_C} \frac{\partial}{\partial \theta} \left\{ \iint (F_{Y|D=1,Z=1,X=x,S=s}(y) \cdot \eta_1(x, s) - F_{Y|D=1,Z=0,X=x,S=s}(y) \cdot \eta_0(x, s)) f(x|s) f(s) dx ds \right\}$$
$$- \left\{ \iint (F_{Y|D=1,Z=1,X=x,S=s}(y) \cdot \eta_1(x, s) - F_{Y|D=1,Z=0,X=x,S=s}(y) \cdot \eta_0(x, s)) f(x|s) f(s) dx ds \right\} \frac{\partial P_C(\theta_0)}{\partial \theta}.$$

Similarly, we have

$$\frac{\partial}{\partial \theta} F_{Y(0)|\mathcal{C}=c}(y; \theta_0)$$
$$= -\frac{1}{P_C} \frac{\partial}{\partial \theta} \left\{ \iint (F_{Y|D=0,Z=1,X=x,S=s}(y) \cdot \eta_1(x, s) - F_{Y|D=0,Z=0,X=x,S=s}(y) \cdot \eta_0(x, s)) f(x|s) f(s) dx ds \right\}$$
$$+ \left\{ \iint (F_{Y|D=0,Z=1,X=x,S=s}(y) \cdot \eta_1(x, s) - F_{Y|D=0,Z=0,X=x,S=s}(y) \cdot \eta_0(x, s)) f(x|s) f(s) dx ds \right\} \frac{\partial P_C(\theta_0)}{\partial \theta}.$$

We choose $\psi(Y, D, Z, X, S)$ as

$$\psi(Y, D, Z, X, S)$$
$$= \left\{ \frac{Z}{\pi_1(S)} \cdot (\mathbb{1}_{\{Y \leq y\}} - \mu_1(y, S, X)) - \frac{1-Z}{\pi_0(S)} \cdot (\mathbb{1}_{\{Y \leq y\}} - \mu_0(y, S, X)) + \mu_1(y, S, X) - \mu_0(y, S, X) \right\} /$$
$$\left\{ \frac{Z}{\pi_1(S)} \cdot (D - \eta_1(S, X)) - \frac{1-Z}{\pi_0(S)} \cdot (D - \eta_0(S, X)) + \eta_1(S, X) - \eta_0(S, X) \right\} - \beta(y).$$

Then, notice that $\psi$ satisfies (33) and that $\psi$ lies in the tangent space $T$ given in (32). Since $\psi$ lies in the tangent space, the variance bound is given by the expected square of $\psi$:

$$\Omega(y) := \mathbb{E}\big[\psi(Y, D, Z, X, S)^2\big]$$

$$= \mathbb{E}\Bigg[\Bigg(\Big\{\frac{Z}{\pi_1(S)} \cdot \big(\mathbb{1}_{\{Y \leq y\}} - \mu_1(y, S, X)\big) - \frac{1-Z}{\pi_0(S)} \cdot \big(\mathbb{1}_{\{Y \leq y\}} - \mu_0(y, S, X)\big) + \mu_1(y, S, X) - \mu_0(y, S, X)\Big\} \Big/$$

$$\Big\{\frac{Z}{\pi_1(S)} \cdot (D - \eta_1(S, X)) - \frac{1-Z}{\pi_0(S)} \cdot (D - \eta_0(S, X)) + \eta_1(S, X) - \eta_0(S, X)\Big\} - \beta(y)\Bigg)^2\Bigg]$$

$$= \Big\{ \mathbb{E}[\pi_1(S_i)\phi_1(y, S_i, \mathcal{D}_i,)\phi_1(y', S_i, \mathcal{D}_i)] + \mathbb{E}[\pi_0(S_i)\phi_0(y, S_i, \mathcal{D}_i)\phi_0(y', S_i, \mathcal{D}_i)]$$

$$+ \mathbb{E}[\xi_i(y)\xi_i(y')] \Big\} \Big/ \big\{ \mathbb{E}[D(1) - D(0)]^2 \big\}$$

This concludes the proof of part (a).

Next, for part (b), under Assumption 5.1, the regression-adjusted estimator defined in Algorithm 1 satisfies the following asymptotic distribution for any given $y \in \mathcal{Y}$:

$$\sqrt{n}\big(\widehat{\beta}(y) - \beta(y)\big) \rightsquigarrow \mathcal{N}(0, \Omega(y)),$$

where $\Omega(y)$ is the semiparametric efficiency bound derived in part (a). This completes the proof of part (b).

$\square$

# D   Inference

We consider two approaches to estimate the standard errors and construct confidence intervals for the regression-adjusted LDTE, $\widehat{\beta}(y)$, at a given threshold $y \in \mathcal{Y}$. Using the asymptotic distribution derived in Theorem 5.2, we can construct a $(1 - \alpha) \times 100\%$ confidence interval for $\widehat{\beta}(y)$ based on a consistent estimator:

$$\left\{ \widehat{\beta}(y) \pm \Phi^{-1}(1 - \alpha/2) \times \sqrt{\widehat{\Omega}(y)}/\sqrt{n} \right\},$$

where $\Phi$ is the standard normal distribution function. For a 95% confidence interval, $\Phi^{-1}(1 - \alpha/2) = 1.96$. The consistent estimator $\widehat{\Omega}(y)$ is given by

$$\widehat{\Omega}(y) := \frac{\frac{1}{n}\sum_{i=1}^{n}\left[ Z_i\widehat{\phi}_1^2(y, S_i, \mathcal{D}_i) + (1 - Z_i)\widehat{\phi}_0^2(y, S_i, \mathcal{D}_i) + \widehat{\xi}_i^2(y) \right]}{\left(\frac{1}{n}\sum_{i=1}^{n}(\Xi_{1,i}^D - \Xi_{0,i}^D)\right)^2}, \quad \text{where}$$

$$\widehat{\phi}_1(y, s, \mathcal{D}_i) := \tilde{\phi}_1(y, s, \mathcal{D}_i) - \frac{1}{n_1(s)}\sum_{j \in I_1(s)}\tilde{\phi}_1(y, s, \mathcal{D}_j),$$

$$\widehat{\phi}_0(y, s, \mathcal{D}_i) := \tilde{\phi}_0(y, s, \mathcal{D}_i) - \frac{1}{n_0(s)}\sum_{j \in I_0(s)}\tilde{\phi}_0(y, s, \mathcal{D}_j),$$

$$\widehat{\xi}_i(y) := \frac{1}{n_1(s)}\sum_{i \in I_1(s)}(\mathbb{1}_{\{Y_i \leq y\}} - \widehat{\beta}(y)D_i) - \frac{1}{n_0(s)}\sum_{i \in I_0(s)}(\mathbb{1}_{\{Y_i \leq y\}} - \widehat{\beta}(y)D_i),$$

$$\tilde{\phi}_1(y, s, \mathcal{D}_i) := \left[\left(1 - \frac{1}{\widehat{\pi}_1(s)}\right)\widehat{\mu}_1(y, s, X_i) - \widehat{\mu}_0(y, s, X_i) + \frac{\mathbb{1}_{\{Y_i \leq y\}}}{\widehat{\pi}_1(s)}\right]$$

$$- \widehat{\beta}(y)\left[\left(1 - \frac{1}{\widehat{\pi}_1(s)}\right)\widehat{\eta}_1(s, X_i) - \widehat{\eta}_0(s, X_i) + \frac{D_i}{\widehat{\pi}_1(s)}\right], \quad \text{and}$$

$$\tilde{\phi}_0(y, s, \mathcal{D}_i) := \left[\left(\frac{1}{\widehat{\pi}_0(s)} - 1\right)\widehat{\mu}_0(y, s, X_i) + \widehat{\mu}_1(y, s, X_i) - \frac{\mathbb{1}_{\{Y_i \leq y\}}}{\widehat{\pi}_0(s)}\right]$$

$$- \widehat{\beta}(y)\left[\left(\frac{1}{\widehat{\pi}_0(s)} - 1\right)\widehat{\eta}_0(s, X_i) + \widehat{\eta}_1(s, X_i) - \frac{D_i}{\widehat{\pi}_0(s)}\right].$$

Second, an alternative method for inference is empirical bootstrap. The procedure is summarized in Algorithm 2.

---

**Algorithm 2** Bootstrap confidence intervals for regression-adjusted LDTE

---

**Input:** Original sample $\{(Y_i, D_i, Z_i, S_i, X_i)\}_{i=1}^n$
**Output:** $(1 - \alpha) \times 100\%$ confidence intervals for the regression-adjusted LDTE

**1.** For each bootstrap iteration $b = 1, \ldots, B$:

**2.** Draw a bootstrap sample of size $n$ with replacement:
$\{(Y_i^b, D_i^b, Z_i^b, S_i^b, X_i^b)\}_{i=1}^n$ from $\{(Y_i, D_i, Z_i, S_i, X_i)\}_{i=1}^n$

**3.** Compute regression-adjusted LDTE $\widehat{\beta}(y)$ given the conditional distribution estimator based on the original sample

**4.** Calculate standard errors $\widehat{\Sigma}(y)$ as the standard deviation of the bootstrapped LDTEs $\{\widehat{\beta}(y)\}_{b=1}^B$,

**5.** Construct the confidence band:

$$\left\{ \widehat{\beta}(y) \pm \Phi^{-1}(1 - \alpha/2) \times \widehat{\Sigma}(y) : y \in \mathcal{Y} \right\},$$

where $\Phi$ is the standard normal distribution function.

---

# E   Additional experimental details

All experiments are run on a Macbook Pro with 36 GB memory and the Apple M3 Pro chip. The code is publicly available at https://github.com/CyberAgentAILab/ldte, and the method can be implemented using the Python library `dte-adj` (https://pypi.org/project/dte-adj/).

Table 4: Pre-treatment covariates included in regression adjustment in Oregon Health Insurance Experiment

| Variable |
| --- |
| Number of ED visits pre-randomization |
| Number of ED visits resulting in a hospitalization, pre-randomization |
| Number of Outpatient ED visits, pre-randomization |
| Number of weekday daytime ED visits, pre-randomization |
| Number of weekend or nighttime ED visits, pre-randomization |
| Number of emergent, non-preventable ED visits, pre-randomization |
| Number of emergent, preventable ED visits, pre-randomization |
| Number of primary care treatable ED visits, pre-randomization |
| Number of non-emergent ED visits, pre-randomization |
| Number of unclassified ED visits, pre-randomization |
| Number of ED visits for chronic conditions, pre-randomization |
| Number of ED visits for injury, pre-randomization |
| Number of ED visits for skin conditions, pre-randomization |
| Number of ED visits for abdominal pain, pre-randomization |
| Number of ED visits for back pain, pre-randomization |
| Number of ED visits for chest pain, pre-randomization |
| Number of ED visits for headache, pre-randomization |
| Number of ED visits for mood disorders, pre-randomization |
| Number of ED visits for psych conditions/substance abuse, pre-randomization |
| Number of ED visits for a high uninsured volume hospital, pre-randomization |
| Number of ED visits for a low uninsured volume hospital, pre-randomization |
| Sum of total charges, pre-randomization |
| Age |
| Gender |
| Health (last 12 months) |
| Education (highest completed) |

