# OpenReview forum: "Beyond the Average: Distributional Causal Inference under Imperfect Compliance"
_NeurIPS.cc/2025/Conference — NeurIPS 2025 poster_

### Official Review · Reviewer_eyve · 2025-06-07

**Clarity:** 4
**Significance:** 3
**Originality:** 3
**Rating:** 5
**Confidence:** 3

**Summary:**

The paper introduces an estimator of the local distributional treatment effect (LDTE) under imperfect compliance to obtain the difference in treatment and outcome distribution for the compliers. The authors introduce a double machine learning approach to estimate the LDTE, show its theoretical properties, including asymptotic normality, and its effectiveness in practical applications.

**Questions:**

- The assumed convergence rates in Assumption 5.1 (i) appear relatively strong in the context of the usual assumption in double-machine learning (where thanks to orthogonalization weaker rates can be assumed usually). Is this just unavoidable in this context?
- Just to clarify, you explain that Assumption 3.1 (i) allows for corss-sectional dependence among treatment statuses. This is because Z_i is left out of the i.i.d. assumption?
- In Equation (2), if I understand correctly, you basically have a ratio of two AIPW estimators. How does this affect the theoretical handling of the estimator as opposed to the usual AIPW estimator? Is this a reason for the convergence rate assumptions?
- I would add () around the citations in line 152

**Ethical Concerns:**

["NO or VERY MINOR ethics concerns only"]

**Final Justification:**

I think this is a great paper with an interesting and useful new method, and solid theory. The authors addressed my questions adequately. Thus, I see no unresolved issue and I am thus deciding to leave my score.

**Limitations:**

Yes

**Quality:**

3

**Strengths And Weaknesses:**

Strengths
-----------------
- Solid paper, introducing an important method with interesting theoretical results. In particular, I like the guarantee of uniform confidence bands.
- Proofs are well written!

Weaknesses
----------------
- While the result in Theorem 5.2. is very nice, it appears to me to not give any insights on how to actually construct confidence intervals, unless the covariance kernel can somehow be estimated. In the appendix you use bootstrap which is usually justified under asymptotic normality. Still I personally would have been more interested in a result justifying the bootstrap or another method to construct the simultaneous confidence bands, than whether beta meets the semiparametric efficiency bound.

---

> ### Author Rebuttal · Authors · 2025-07-31
>
> We appreciate your careful reading of our paper and your insightful questions regarding the technical details. Your thoughtful comments are both encouraging and constructive. Below, we address each of the issues you have kindly raised.
>
> **Bootstrap validity**
>
> Due to space limitations, we do not discuss the bootstrap procedure and its validity in our paper. To our knowledge, we can establish formal bootstrap validity and explain both point-wise and uniform bootstrap procedures. In our revision, we will add several paragraphs regarding bootstrap methodology. To construct simultaneous confidence bands, we can use test inversion with test statistics such as the Kolmogorov-Smirnov and Cramér-von Mises statistics. The bootstrap approximates the distribution of the test statistic under the null hypothesis, and the simultaneous $(1-\alpha)$ confidence band is obtained by inverting the acceptance region of the test, given some significance level $\alpha$.
>
>
> **Convergence rates in Assumption 5.1 (i)**
>
> You are right that in standard double machine learning for mean parameters, orthogonalization typically allows for weaker convergence rate conditions—e.g., $o(n^{-1/4})$. However, in our setting we estimate function-valued parameters indexed by y, and the moment conditions are indexed by strata, which introduces additional technical challenges. The conditions in Assumption 5.1(i) control the stratum-specific differences in nuisance estimation errors across treatment arms, which affect both numerator and denominator of our ratio estimator. We believe these conditions could be relaxed under further structural or sparsity assumptions (e.g., by controlling empirical process terms more carefully).
>
> **Assumption 3.1(i) and Cross-Sectional Dependence**
>
> Yes, that’s correct. Assumption 3.1(i) treats the potential outcomes, compliance types, strata, and covariates as i.i.d., but intentionally excludes $Z_i$ to allow for cross-sectional dependence in treatment assignments. This reflects covariate-adaptive randomization (CAR) designs, where assignments are correlated across units.
>
> **Ratio of AIPW estimators**
>
> Yes, you are correct that our estimator can be viewed as a ratio of two AIPW estimators. More formally, it can be interpreted as solving a system of two Neyman-orthogonal moment conditions: one for the numerator, which captures the distributional intent-to-treat effect, and one for the denominator, representing the first-stage effect of the instrument on treatment uptake. The overall estimator inherits orthogonality from this moment system, ensuring robustness to first-order estimation errors in the nuisance functions.
>
> Our setup parallels that of Chernozhukov et al. (2018), who analyze the estimation of the local average treatment effect (LATE) using double machine learning under a similar orthogonalization framework. Finally, we note that the convergence rate condition in Assumption 5.1(i) is not driven by the ratio form of the estimator, but rather arises from the stratified structure of the randomization and the need to control imbalance in nuisance estimation across treatment groups within strata.
>
> **Citation formats**
>
> We appreciate your suggestion regarding the citation format. We will revise the citations to include parentheses as recommended and ensure consistency throughout the manuscript.
>
> **Final Remarks**
>
> We are grateful for your insightful and constructive feedback. Your comments have helped us clarify subtle theoretical points and identify opportunities for sharpening the presentation. We will incorporate all suggested revisions and note bootstrap validity and potential rate relaxations as important directions for future work.

---

> > ### Comment · Reviewer_eyve · 2025-08-05
> >
> > Thanks for these detailed comments, I am happy to leave my positive score!

---

> > > ### Author Response · Authors · 2025-08-05
> > > **Acknowledgement**
> > >
> > > I appreciate your thoughtful response. Constructive discussions like this greatly improve the quality of our work

---

### Official Review · Reviewer_Nyy9 · 2025-06-26

**Clarity:** 2
**Significance:** 3
**Originality:** 3
**Rating:** 4
**Confidence:** 2

**Summary:**

This paper introduces an estimator for the local distributional treatment effect, which is an instrumental-variable-type estimator for the causal effect among compliers. They prove asymptotic properties of the estimator and empirically argue using synthetic and real data that the estimator is more effective than baselines.

**Questions:**

- clarify the contribution beyond previous work
- correct Fig 4 so that the L and R parts are different
- clarify how I should interpret causal graph in Fig 1 and how it interacts with the definitions of \eta and \mu

**Ethical Concerns:**

["NO or VERY MINOR ethics concerns only"]

**Final Justification:**

After looking at the other reviewers points and the rebuttal, I'm increasing to a borderline accept - I didn't grasp all of the contributions of this paper in the original reading.

**Quality:**

3

**Strengths And Weaknesses:**

Strengths:
- while I do have some confusions outlined below, I think the contribution here is probably theoretically solid and well-scoped
- experiments are mostly clear and compelling

Weaknesses:
- could use some more clarity on exactly what the contribution here is - is it just adding a distributional component onto existing IV estimators for complier effects? or something further. I think it's more but have trouble totally figuring it out, the distributional component by itself seems like probably not quite enough novelty but I could be wrong. Lemma 3.2 would be a nice place to do some of this clarification but anywhere is fine
- confused about a couple fundamental pieces here: the authors define \eta_z = E[D | Z, S, X] on L193. However according to the causal graph in Fig 1 it looks like D should be independent of X, S given Z - why is this not just E[D | Z] in that case? Similarly I wonder about the equation below on L194 - I would assume from Fig 1 that Y should be independent of X, S given Z
- Algo 1: should clarify what exactly M is trained on, inputs + outputs. I don't see this notation elsewhere
- Assumption 5.1: (I) I find it confusing why we need a condition on the difference but not on the individual components, (ii) the < \infty part of this is confusing to me as well, doesn't this hold trivially for any F bounded above by 1? which \mu_z should be since it is an Expectation of an indicator function
- Theorem 5.2 - I may just not have the theory background here so apologies but I don't really understand what it means for this to converge to a GP. Don't we want it to be decreasing in some function of n?
- Fig 4: your left and right plots look identical to me, I'm assuming this is an error

---

> ### Author Rebuttal · Authors · 2025-07-31
>
> We thank the reviewer for the thoughtful feedback and for recognizing the strength of our theoretical contributions and empirical evaluation. We appreciate the opportunity to clarify the technical aspects and novelty of our work, and we address each of your comments below.
>
> **Clarifying the Contribution Beyond Prior Work**
>
> Our contribution goes beyond simply applying distributional analysis to existing IV estimators. Specifically, we provide:
> 1. A new regression-adjusted estimator for the local distributional treatment effect (LDTE) that is valid under covariate-adaptive randomization (CAR) and imperfect compliance. Prior work (e.g., Abadie, 2002) does not accommodate CAR.
> 2. Theoretical guarantees under CAR designs, including asymptotic normality and efficiency, extending beyond i.i.d. designs.
> 3. Neyman-orthogonal moment conditions that enable robust estimation with high-dimensional covariates using machine learning methods.
> 4. A unified framework that applies to continuous, discrete, and mixed outcomes under a wide range of randomization designs (stratified and unstratified).
> We agree that Lemma 3.2 is a natural place to clarify the identification strategy, and we will revise the surrounding text to better articulate our contribution.
>
> **Clarifying the Role of Covariates in $\eta_z$ and $\mu_z$**
>
> You raise an important question regarding the conditional expectations. While the graph in Figure 1 shows arrows from $X$ and $S$ to $Z$, and from $Z$ to both $D$ and $Y$, the lack of arrows from $X$, $S$ to $D$ or $Y$ does not imply that $D$ and $Y$ are conditionally independent of $X$ and $S$ given $Z$. This is because the graph encodes the randomization mechanism and exclusion restrictions, but not the full dependence structure of post-randomization variables (like $D$ and $Y$) on pre-treatment covariates. In particular:
> * Even under perfect randomization, the realized treatment $D$ may still vary with $X$ and $S$ due to heterogeneous compliance behavior (e.g., older individuals may comply more).
> * Likewise, $Y$ may depend on $X$ and $S$ through baseline heterogeneity in potential outcomes.
> Hence, conditioning on X and S improves estimation efficiency, even though $Z$ is randomized. This is standard in regression-adjusted causal inference literature (e.g., Lin 2013).
> We will revise Figure 1 and the accompanying text to clarify that the arrows reflect assignment mechanisms, not conditional independence assumptions post-randomization.
>
> **Algorithm 1 – Clarifying Model $\mathcal{M}$**
>
> Thank you for pointing this out. In Algorithm 1, the supervised model $\mathcal{M}$ is used to estimate the nuisance functions $\eta_z$ and $\mu_z$. Each model $\mathcal{M}$ is trained on observations with $Z=z$ and $S=s$, using $X$ as features and either $1\\{Y\leq y\\}$ or $D$ as targets. We will revise the algorithm to clearly specify the inputs and outputs to the model for each step.
>
> **Assumption 5.1 – Clarifying Conditions**
>
> Assumption 5.1(i) ensures that the average estimation errors of the nuisance functions are balanced across treatment groups within each stratum. It is required due to the structure of our estimator, which relies on differences in stratified group-level averages.
>
> (ii) Thank you for pointing this out. Assumption 5.1(ii) is used to control the complexity of the nuisance function class $\mathcal{F}_z$ in our empirical process analysis. In our setting, each $\mu_z(y, s, x)$ is a conditional CDF and thus takes values in $[0,1]$, making the envelope function uniformly bounded. Hence, the moment condition holds trivially for all $q>2$. We will clarify this in the revised version.
>
> **Theorem 5.2 – Interpretation of Convergence to a GP**
>
> This theorem establishes weak convergence of the estimator process to a mean-zero Gaussian process (GP) indexed by $y \in \mathcal{Y}$. This result is the functional analog of the classical central limit theorem: rather than proving pointwise asymptotic normality at a fixed $y$, it shows convergence of the entire function-valued estimator to a Gaussian process. This is useful to quantify uncertainty surrounding the estimation of functions using random data.
>
> For your question: "Don't we want it to be decreasing in some function of n?", yes, in some sense. Notice that $\sqrt{n}(\hat{\beta}(y) - \beta(y)) = O_p(1)$ implies that $\hat{\beta}(y) - \beta(y) = O_p(n^{-1/2})$, or that $\hat{\beta}(y)$ converges to $\beta(y)$ in probability as $n$ goes to infinity. Thus, your intuition is correct, and furthermore the theorem provides exactly how estimation error behaves approximately when $n$ goes to infinity.
>
> **Figure 4**
>
> Thank you for raising this point. Figure 4 displays both unadjusted (left panels) and regression-adjusted (right panels) estimates of the LDTE and LPTE from the Oregon Health Insurance Experiment. While the point estimates appear visually similar across panels, the key difference lies in the width of the confidence intervals. As detailed in the main text, regression adjustment using gradient boosting reduces the standard errors by approximately 10–20% for LDTE and 5.5–10% for LPTE. These efficiency gains are not always visually prominent, but they are quantitatively meaningful. We will clarify this further in the figure caption and ensure the visual distinction is easier to interpret in the final version.
>
> **Final Remarks**
>
> We again thank the reviewer for the thoughtful and constructive feedback. We believe the clarifications and revisions outlined above will address the key concerns and help make the paper more accessible and precise.

---

> > ### Comment · Reviewer_Nyy9 · 2025-08-05
> >
> > Thanks for the rebuttal. A couple helpful points in here and inclined to bump my score but will take a look at the paper first. Quick note that I do see the two halves of Fig 4 are technically non-identical now but my point remains that any distinction you're trying to communicate is not coming across there

---

> ### Author Response · Authors · 2025-08-06
> **Acknowledgement**
>
> We appreciate your positive evaluation and valuable comments. In our revision, we will carefully examine how to present our empirical results, particularly Figure 4, more clearly to enhance communication with readers. We will also incorporate all of your suggestions to improve the revised paper. Once again, we are truly grateful for your thoughtful guidance.

---

### Official Review · Reviewer_cLuE · 2025-07-02

**Clarity:** 2
**Significance:** 2
**Originality:** 2
**Rating:** 4
**Confidence:** 3

**Summary:**

This paper proposes a framework for estimating the local distributional treatment effect for experiments with non-compliance for covariate-dependent randomization scenarios. For that, they provide a distributional regression-adjusted estimator. The authors derive the asymptotic properties of the estimator and show that, given additional assumptions, it can achieve the semi-parametric efficiency bound.  Then they provide results for experiments on synthetic and real-world data.

**Questions:**

- l.56: What is meant by “traditional i.i.d. framework” in this context? Does this refer to normal randomized treatment assignment independent of covariates?
- The equation after l.127 holds only under consistency assumptions for potential outcomes of Y and potential outcomes of D, right? I think this should be stated and maybe, together with the formal description of the exclusion assumption, added to assumptions 3.1
- Eq. (1): The notation here (and later on) seems to me a bit untypical. Is there any reason to express the CDFs here as expectations over indicators instead of using the probability as after l.164? Also, in the denominator, for binary D, I would usually expect $\mathbb{P}(D=1|...)$ instead of $\mathbb{E}[D|...]$
- l. 289: What is the Local Probability Treatment Effect, and why is it only introduced here for the evaluation?

**Ethical Concerns:**

["NO or VERY MINOR ethics concerns only"]

**Final Justification:**

Most of my concerns have been addressed in the rebuttal. While I think the empirical evaluation could still be improved clearly, I understand that the major contribution lies in introducing the framework for the new setting and the provided theoretical guarantees. Since in typical causal inference papers, the experiments are more like a proof of concept due to the problem that counteractuals/treatment effects cannot be observed directly in the real world, I currently tend to vote for acceptance of this paper, given that the unclarities and disambiguitites which I mentioned in my original review will be resolved for the final version as discussed in the rebuttal.

**Limitations:**

Major limitations are mentioned in the paper.

**Quality:**

2

**Strengths And Weaknesses:**

# Strengths:
- Extending the LDTE for the CAR setting with non-compliance seems to be novel and is of real-world relevance since non-compliance exists in various applications from A/B tests in marketing to clinical trials in medicine.
- The paper provides proper theoretical proofs for most of its statements in the main paper and in the Appendix.
- Providing insights on real-world data with the Oregon Health Insurance Experiment is a suitable application for their method and goes beyond only synthetic evaluations as often seen in causal inference research.
# Weaknesses:
- Some parts of the paper are only described vaguely, not sufficiently supported by additional evidence, or are hard to understand. Examples in the following:
    -  ll.205: Here, the first-order sensitivity of the proposed estimator should be shown
    - ll: 208: “To further improve robustness, we incorporate cross-fitting with L folds (L > 1) as proposed by Chernozhukov et al. (2018).” I think this really vague. The cross-fitting is necessary to ensure the theoretical guarantees of the estimators, which should be listed above.
    - The authors state that their method is “applicable to continuous, discrete, and mixed discrete-continuous outcomes”. However, in Algorithm 1, it is necessary to iterate through all levels of $\mathcal{Y}$, which should be infeasible for continuous outcomes, or requires strong simplifications such as stratifications for which guaranteeing identifiability is not always straightforward. Overall, since this statement also appears in the abstract, I think evidence for that should be provided.
- The empirical evaluation could be improved clearly. Here, only ML adjustment and linear adjustments are compared. No baselines are added (e.g., comparing it to a naive ITT estimate of the DTE, etc.). If none are applicable or not useful, I would strongly suggest discussing this in the paper. Also, for the synthetic data, what values for $y$ are evaluated?
- The contribution of the method is limited, as it mostly seems to extend Abadie, A. (2002) to the CAR setting.

Minor:
- I think the challenges of CAR and the connection to regression adjustment could be described more clearly in the introduction to give a better intuition of their contribution extending over simple randomization.
- All equations should be numbered.
- Plot descriptions should be standalone and better explain the plots and their implications.

---

> ### Author Rebuttal · Authors · 2025-07-31
>
> We thank the reviewer for the careful reading and valuable comments. We appreciate that you found our work on noncompliance to be novel and practically relevant, and that you recognized the strength of both our theoretical contributions and real-world application.
>
> Below we address your concerns in detail and outline the revisions we will make to improve clarity and completeness.
>
> **Line 205 – First-order sensitivity**
>
> We agree that the first-order sensitivity of the estimator should be more explicitly addressed. Our estimator satisfies Neyman orthogonality, which implies it is first-order insensitive to estimation errors in nuisance functions (i.e., the conditional outcome distributions). We will revise the main text to explicitly highlight this link and refer to the score-based derivation in the Appendix.
>
> **Line 208 – Cross-fitting and its necessity**
>
> Thank you for pointing this out. To clarify: cross-fitting is not required for the validity of our asymptotic theory under standard regularity conditions, including when nuisance functions are estimated at sufficiently fast rates. Our theoretical results do not rely on sample-splitting. However, we incorporate cross-fitting in our implementation because it is known to improve finite-sample performance, especially when using flexible machine learning methods for nuisance estimation (Chernozhukov et al., 2018)
>
> **Line 56 – Traditional i.i.d. framework**
>
> By “traditional i.i.d. framework,” we refer to experimental designs assuming that treatment assignment is independent and identically distributed, i.e., independent of covariates and not adaptive. This contrasts with CAR, where assignments depend on strata or covariates. We will clarify this in the text.
>
> **Outcome and treatment equations and Assumption 3.1**
>
> Yes, the identification result following line 127 relies on a no interference assumption for both outcomes $Y$ and treatment receipt $D$. We will incorporate this formally into Assumption 3.1. Additionally, while the exclusion restriction is currently stated informally, we agree it should be presented more rigorously and will revise Assumption 3.1 accordingly in the next version.
>
> **Equation (1) – Notation for CDFs and expectations**
>
> We chose to express the empirical CDFs using expectations over indicators (e.g., $E[1\\{Y\leq y\\}∣\cdot]$) for ease of estimation and because this representation directly connects to the moment conditions used in our estimator. For binary random variable $D \in\\{0,1\\}$, $E[D∣\cdot]$ and $P(D=1∣\cdot)$ are equivalent, and we will clarify this in the text to avoid confusion.
>
> **Applicability to continuous outcomes**
>
> We appreciate the chance to clarify this important point. In Algorithm 1, the iteration over $y \in \mathcal{Y}$ is conceptual and intended to cover a grid of outcome values in practice. This is standard in distributional regression frameworks (e.g., Chernozhukov et al., 2013). We emphasize that the grid spans the support of $Y$ and does not involve looping over individual outcome observations. We will revise the paper to make this distinction explicit and clarify the applicability to continuous outcomes.
>
> **Empirical Evaluation and Baselines**
>
> Thank you for this suggestion. In both our synthetic data experiments and real-world data application, we compare the regression-adjusted estimators—both linear and machine-learning-based—to the unadjusted estimator, which serves as our baseline. We will clarify this comparison more explicitly in the revised version of the paper and discuss its role as a baseline.
>
> **What values of y are used in synthetic evaluation?**
>
> We evaluate the LDTE on quantiles $y\in \\{0.1,…,0.9 \\}$ of the outcome distribution in the simulation study. This grid was chosen to provide interpretable coverage across the distribution. We will clarify this in the experimental section.
>
> **Clarifying LPTE (line 289)**
>
> We appreciate the reviewer noting this. The Local Probability Treatment Effect (LPTE) estimates the difference in point mass (probabilities) at each discrete outcome level, analogous to a probability mass function comparison for compliers. We introduced it to complement the Local Distributional Treatment Effect (LDTE) for discrete outcomes (e.g., count data such as ED visits). We will add a formal definition earlier in the paper and clarify its distinction from the LDTE.
>
> **Positioning and Contribution**
>
> While our identification strategy builds on Abadie (2002), our estimator significantly extends that work in several ways:
> 1. We accommodate covariate-adaptive randomization (CAR), whereas Abadie (2002) assumes i.i.d. assignment.
> 2. We incorporate regression adjustment using high-dimensional covariates, enhancing precision.
> 3. We provide semi-parametric efficiency bounds and a doubly robust estimator.
> We will emphasize this distinction more clearly in both the introduction and related work sections. Additionally, it's worth noting that Abadie (2002) focuses on a hypothesis test for equality of distributions, which is fundamentally different from our goal of estimating and conducting inference on distributional treatment effects.
>
> **Minor Comments**
>
> * We will improve the introduction to better explain the challenges posed by CAR and its connection to regression adjustment.
> * We will number all equations in the revised version.
> * We will revise plot captions to be more self-contained and interpretative.
>
> **Final Remarks**
>
> We thank the reviewer again for these detailed and constructive comments. We believe that with the suggested clarifications and additional comparisons, the revised paper will more clearly communicate both its theoretical and practical contributions.

---

> > ### Comment · Reviewer_cLuE · 2025-08-05
> >
> > Thank you for your rebuttal! Most of my concerns have been addressed. While I think the empirical evaluation could still be improved clearly, I understand that the major contribution lies in introducing the framework for the new setting and the provided theoretical guarantees. Since in typical causal inference papers, the experiments are more like a proof of concept due to the problem that counteractuals/treatment effects cannot be observed directly in the real world, I currently tend to vote for acceptance of this paper, given that the unclarities and disambiguitites will be resolved for the final version as discussed in the rebuttal.

---

> ### Author Response · Authors · 2025-08-06
> **Acknowledgement**
>
> We are grateful for your positive assessment and constructive comments, which will be invaluable in helping us revise our paper. As you noted, we will make every effort to address any ambiguities and unclear elements, particularly in our empirical results. We sincerely appreciate the opportunity to receive your valuable feedback.

---

> > ### Author Response · Authors · 2025-08-07
> > **Acknowledgement**
> >
> > We appreciate your positive assessment and constructive comments, which will be invaluable in helping us revise our paper.

---

### Official Review · Reviewer_yo62 · 2025-07-05

**Clarity:** 3
**Significance:** 3
**Originality:** 3
**Rating:** 4
**Confidence:** 3

**Summary:**

In this paper, the author studies the problem of distributional effect estimation under imperfect compliance. In the imperfect compliance setting, people can choose to move to different treatments or not comply with the treatment they were assigned. While there have been many works on treatment effect estimation under imperfect compliance, there is little known work on distributional effect estimation, which measures the distributional effect between treatment and control groups. In this paper, the authors introduce a regression-adjusted estimator for local distributional treatment effects (LDTEs) with covariate adaptive randomization (CAR). They provide asymptotic properties, including semi-parametric efficiency, and demonstrate the effectiveness of the estimator on real and synthetic data sets. The real data set is the Oregon Health Insurance Experiment, where compliance is only 50-60%.

**Questions:**

Can the authors comment on **non-asymptotic** properties of the regression estimator proposed? How exactly would it depend on sampling strategy.

**Ethical Concerns:**

["NO or VERY MINOR ethics concerns only"]

**Final Justification:**

I want to thank the authors for their rebuttal response. I am leaning towards borderline accept.

**Quality:**

3

**Strengths And Weaknesses:**

**Strengths**

The authors address an important question of distributional effect estimation (quantile treatment effects). While most work has focused on mean estimation, there has been a growing body of work for the more challenging distributional estimation estimation problem. This work contributes to that body, under an important realistic setting of imperfect compliance.

The paper is the first to combine regression adjustment and instrumental variable methods for estimating full outcome distributions under CAR and imperfect compliance.

**Weaknesses**

It is not entirely clear why a strata based conditional estimator was proposed. The authors could have easily studied the same setting without introducing strata, and just using covariates. It would also be interesting to see if there is any empirical (or theoretical) performance gains obtained by stratifying.

It is also not clear from the paper, why balancing based approaches fail for distributional effect estimation (among the complied). The authors propose a regression based estimator.

---

> ### Author Rebuttal · Authors · 2025-07-31
>
> We sincerely thank the reviewer for the thoughtful and constructive feedback. We are encouraged by the positive assessment of our work’s significance and originality, and we are grateful for the opportunity to clarify our contributions and address the questions raised.
>
> **On the Use of Stratification**
>
> While a strata-based formulation may appear more restrictive than models based purely on covariates, our framework in fact generalizes both stratified and unstratified experimental designs. Specifically, the case of no stratification corresponds to the special case where there is a single stratum (S=1), which is explicitly covered by our setup and results. Thus, our identification, estimation procedures, and semiparametric efficiency bounds apply broadly—including to simple random assignments and covariate-adaptive designs.
> Importantly, our framework also allows for covariate adjustment using variables beyond those used for stratification, further enhancing flexibility. In spirit, both stratification and covariate adjustment are tools to reduce the variance of treatment effect estimators by incorporating available information—either through design or analysis (see, e.g., Lin, 2013; Imbens & Rubin, 2015). We will revise the paper to make this generality and motivation clearer.
>
> **On Regression vs. Balancing-Based Approaches**
>
> Thank you for raising this important point. Our empirical (or unadjusted) estimator can indeed be interpreted as a balancing-based method, as it involves inverse-propensity weighting (IPW) components —featuring weighted averages in both the numerator and denominator. This estimator is unbiased and consistent for the distributional treatment effect among compliers.
> However, balancing alone does not fully exploit the available information in covariates X. Our proposed regression-based estimator incorporates additional covariate adjustment, which improves efficiency (i.e., reduces variance) relative to the IPW-only approach. We will clarify this connection in the revised manuscript and highlight the advantages of regression adjustment in this context.
>
> **Non-Asymptotic Behavior and Sampling Strategy**
>
> We appreciate your interest in the finite-sample performance of our estimator. As discussed in Section 6, our simulations show that both linear and machine-learning-based regression adjustments substantially improve RMSE and confidence interval length compared to the unadjusted estimator, with gains becoming more pronounced as the sample size increases.
> The non-asymptotic behavior of such estimators depends on several factors, including the complexity of the nuisance functions, the use of regularization, and cross-fitting strategies. While our theoretical analysis focuses on asymptotic properties, our simulation results provide strong empirical evidence that the proposed methods perform well even in moderately sized samples.
> We also note that recent work has begun to establish non-asymptotic guarantees theoretically in related settings, particularly for average treatment effect estimators using regression adjustments and propensity score corrections. Notably, Su et al. (2023a) propose a decorrelation-based method that achieves finite-sample performance guarantees for general regression adjustments, and Su et al. (2023b) provide high-dimensional bias correction and analysis for propensity-score-based estimators:
> * Su, F., Mou, W., Ding, P., & Wainwright, M. J. (2023a). A decorrelation method for general regression adjustment in randomized experiments. arXiv:2311.10076.
> * Su, F., Mou, W., Ding, P., & Wainwright, M. J. (2023b). When is the estimated propensity score better? High-dimensional analysis and bias correction. arXiv:2303.17102.
>
> We agree that extending such non-asymptotic theory to distributional treatment effect estimation would be a valuable and promising direction for future research.
>
> **Final Remarks**
>
> We are grateful for your feedback and will incorporate the suggested clarifications into the final version of the paper. We believe these additions will make the contributions and scope of our work more transparent.

---

### Decision · Program_Chairs · 2025-09-17

**Decision:**

Accept (poster)

**Comment:**

The authors propose a new approach for estimating distributional (quantile) treatment effects in randomized experiments with covariate-adaptive randomization (CAR), which allows treatment assignment to be correlated across units, and noncompliance (e.g., participants do not adhere to their assigned treatment).  While distributional treatment effects and CAR have been studied separately, this is the first work to integrate these areas of causal inference.  The authors propose a doubly robust, sample-efficient regression-adjusted estimator for the local distributional treatment effect (for compliers, across treatment arms).  They prove a semiparametric efficiency bound for this estimator and show theoretically and empirically that the variance is lower for the regression-adjusted as compared to an unadjusted estimator.  Finite-sample performance is evaluated for a simulation study and for real-world data from the Oregon Health Insurance Experiment.

Strengths:
* Both distributional treatment effects and noncompliance are important areas of causal inference, and this paper integrates the two in a rigorous, correct, and compelling way.
* The authors provide convincing evidence (theory, simulation, and an experiment on real-world data) that the regression-adjusted estimator outperforms the unadjusted estimator.
* The paper is generally clear and well written.
* Theoretical results are interesting, and experiments are clear and compelling.  I particularly appreciated the use of real-world Oregon Health Insurance Experiment data rather than just simulations.

Weaknesses:
* The main theoretical results are not particularly surprising - e.g., the proposed estimator can be written as the ratio of two augmented inverse probability weighted (AIPW) estimates for change in outcome distribution and change in treatment uptake respectively, and one might expect covariate adjustment to improve statistical efficiency and reduce variance. (Nevertheless, the results are useful and derived in a rigorous and convincing manner.)
* Various aspects of the original submission needed clarification, though these were resolved adequately in discussion.

Reason for accept/reject decision: This is a solid paper which makes a useful and important contribution to the causal inference ML literature, addressing the combination of distributional treatment effects and CAR/noncompliance for the first time.  I believe the paper should be accepted, but would hesitate to award a spotlight or oral given that the results are strong and useful but not earth-shattering.

Discussion: There was a robust discussion between the authors and reviewers, in which the authors were able to clarify several points which were initially unclear to the reviewers, for example (i) making it clear that the current work incorporates CAR while prior work of Abadie (2002) assumes i.i.d. treatment assignment; (ii) defining the local probability treatment effect; (iii) clarifying specific aspects of the model and assumptions; and (iv) explaining the convergence rate. It is important for these additional discussion points to be incorporated into the final version of the paper, and I believe that we can rely on the authors to do so, particularly given that this will further increase the accessibility and impact of the paper. No major issues, nor need for substantial additional experiments, were identified during the discussion, and the reviewers were generally satisfied with the authors' responses, rating the paper between borderline accept and full accept after discussion.